# Widespread flooding dynamics under climate change: characterising floods using grid-based hydrological modelling and regional climate projections

Adam Griffin[1], Alison L. Kay[1], Paul Sayers[2], Victoria Bell[1], Elizabeth Stewart[1], Sam Carr[2]

[1]UK Centre for Ecology & Hydrology, Wallingford, Oxfordshire, OX10 8BB, UK.
[2]Sayers and Partners, Watlington, Oxfordshire, OX49 5PY, UK.

*Correspondence to*: Adam Griffin (adagri@ceh.ac.uk)

**Abstract.** An event-based approach has been used to explore the potential effects of climate change on the spatial and temporal coherence of widespread flood events in Great Britain. Time series of daily mean river flow were generated using a gridded national-scale hydrological model (Grid-to-Grid) driven by a 12-member ensemble of regional climate projections from UK Climate Projections 2018 (UKCP18), for 30-year baseline (1980-2010) and future (2050-2080) time-slices. From these, sets of widespread extreme events were extracted. The question of what defines a "widespread flood event" is discussed; here these were defined as exceeding an at-site 99.5th percentile (equivalent to two days per year) simultaneously over an area of at least 20 km², with a maximum duration of 14 days. This resulted in a set of 14,400 widespread events: approximately 20 events per year, per ensemble member, per time-slice. Overall, results have shown that events are more temporally concentrated in winter in the future time-slice compared to the baseline. Distributions of event area were similar in both time-slices, but the distribution of at-site return periods showed some heavier tails in the future time-slice. Such information could be useful for adaptation planning and risk management for floods under climate change, but the potential future changes have to be interpreted in the context of some differences in event characteristics between the baseline climate projection-driven model runs and an observation-driven model run. While the focus here is Great Britain, the methods and analyses described could be applied to other regions with hydrological models and climate projections of appropriate resolution.

## 1    Introduction

River floods are a major natural hazard globally, occurring at a range of spatial and temporal scales (Kundzewicz et al. 2019). Managing flood risk (the combination of hazard with exposure and vulnerability) is important, both from an economic and social perspective. Flood prediction, and more generally flood frequency estimation, is crucial to mitigating these hazards to reduce impact. Flood frequency estimation is often carried out on a single-site basis, computing the frequency of floods at specific locations in isolation. However, the management of flood risk on a regional or national basis requires an understanding of how likely it is that multiple locations will experience floods at the same time. Widespread flooding presents a huge challenge for communities and emergency response services and has long-lasting impacts, as demonstrated by the extensive flooding experienced in North-West England as a consequence of Storm Desmond and Storm Frank in winter 2015/2016

(Barker et al., 2016), Hurricane Katrina in 2005 in the USA (Irish et al., 2014), flooding in Kerala, India, in 2018 (Vishnu et al., 2019) and the flooding in central Europe in July 2021 (Mohr et al., 2021).

One approach to risk quantification is catastrophe modelling (CAT modelling), which is used in the insurance industry to assess annual average losses. CAT modelling typically makes use of three components: property data, stochastic hazard event sets and a relationship between magnitude of hazard and the expected loss for each property (Grossi and Kunreuther, 2005). The present work focuses on the second component: developing a set of widespread flood events, here characterised by river flow and the probability of exceeding that flow. Simply making use of observed widespread events typically does not provide enough data to reliably determine hazard probability. Therefore, developing a larger set of events for analysis is desirable to improve the uncertainty of risk estimates, particularly for events which have a return period (or average recurrence interval) greater than the length of observed records. For example, return periods as long as 1 in 200 years are often used as the design standard for large-scale engineering projects. This can involve making use of hydrological models driven by large ensembles of data from climate models (ensembles of model runs using perturbed initial conditions and/or parameter sets) over a shorter time period (Kelder et al., 2020), or through predominantly stochastic event-based models (Filipova et al., 2019).

Climate change affects flow regimes globally (Jiménez Cisneros, 2015), and studies suggest that flooding in some parts of Europe and the UK could become more frequent and severe in future (Thober et al. 2018, Collet et al., 2018). Spatial coherence of flooding events – whether flood timings at different locations have become more correlated – is of key interest to national-scale actions to mitigate the associated loss. The dependence structure of river flow has been analysed on a Europe-wide scape (Berghuijs et al., 2019) and for the United States (Brunner et al., 2020), focusing on synchrony of events within a given range. The UK's Third Climate Change Risk Assessment (CCRA3) included work which analysed the changes in risk caused by possible changes in flood dynamics (Sayers et al., 2020). CCRA3 adds to the breadth of guidance that has been developed for policymakers and water managers to try and account for such changes (Reynard et al., 2017).

This paper makes use of a grid-based hydrological model for Great Britain (GB) and the most recent set of regional climate projections for the UK to generate two sets of over 7000 hazard events for the recent past (1980-2010) and the future (2050-2080). The question of what defines a "widespread flood event" is discussed, and differences between events in terms of extent, likelihood and duration are analysed in the context of possible changes in the spatio-temporal structure of widespread events in the future. Often flooding is considered on a site-by-site or regionally summarised fashion, particularly when looking into projections of the future. This paper hopes to show the benefits of considering widespread flooding events over a large area using grid-based, rather than catchment-based, hydrological modelling to expand our knowledge of the extent of possible flooding events. While the focus here is GB, the methods and analyses could be applied to other areas where appropriate hydrological models and high resolution climate projections are available. Note that, within the context of flood frequency, this paper refers to floods or flooding events, although in reality many of these will be merely high flows that do not exceed bankfull.

## 2    Data

Hydrological model runs are performed using both observation-based data and baseline and future climate projection data. The model (Section 3.1) requires gridded time-series of precipitation and potential evaporation (PE), plus temperature for the optional snow module.

### 2.1    Observation-based driving data

The observation-based run uses daily 1km precipitation from CEH-GEAR (Tanguy et al., 2019), monthly 40km short grass PE from MORECS (Hough and Jones, 1997) copied down to 1km, and daily 1km minimum and maximum temperatures (Met Office et al., 2019). Precipitation was subdivided uniformly through the day, PE was subdivided uniformly through the month, and temperature varied sinusoidally between the daily extremes (as Kay et al. 2023). Data are applied for 1980-2010 (the same period as the climate model baseline, Section 2.2).

### 2.2    Climate projections

UK Climate Projections 2018 (UKCP18) provides information on potential changes in a range of climate variables over the 21st century, via a number of different products (Murphy et al., 2018). The projections have previously been used to assess how river flows in the UK may differ in the future due to climate change (Kay 2021; Kay *et al.*, 2021; Lane and Kay 2021). The UKCP18 Regional Projections (Met Office Hadley Centre, 2018b) comprise a 12-member perturbed parameter ensemble (PPE) of the Hadley Centre ~12km Regional Climate Model (RCM), nested in an equivalent PPE of their ~60km Global Climate Model (GCM). The ensemble covers the period December 1980 to November 2080 under an RCP8.5 emissions scenario (Representative Concentration Pathway) (Riahi et al., 2011). The 12 ensemble members are numbered from 01 to 15, where 01 uses the "standard" parameterization of the Hadley Centre RCM, and ensemble members 02, 03 and 14 are not available. The data are available re-projected from the native climate model grid to a 12 km grid aligned with the GB national grid, for a synthetic 360-day year (30 days per month). The re-projected daily precipitation and daily minimum and maximum temperatures are used here, along with daily 12km PE calculated from other meteorological variables available from the RCM ensemble (Robinson et al. 2021, 2023). The precipitation data are downscaled to 1km using patterns of standard average annual precipitation (Kay et al. 2023), the PE data are copied down to 1km, and the temperature data are downscaled to 1km using elevation data and a lapse rate (as Kay 2021). The data are then temporally-distributed through the day as for the observed data (Section 2.1). Data are applied for 30-year baseline (1980-2010) and future (2050-2080) time-slices.

In common with all climate model data, there are biases in the UKCP18 Regional data (Murphy et al. 2018 Section 4.4), including a tendency to overestimate winter precipitation rates and fraction of wet days. Often, statistical methods are applied to adjust distributions of climate model data to match those from observed data. However, there are many issues and assumptions inherent in such methods (not least the assumption that the same 'bias' seen in the baseline period will apply in any future period) and they can potentially introduce artefacts into the data (e.g. Ehret et al. 2012, Maraun et al. 2017). A

previous study showed that applying a simple monthly mean correction to precipitation led to flood peaks that had a greater tendency to underestimation, compared to the use of raw precipitation data (Kay 2022, Supplementary Figure 1), thus it was decided to use the raw precipitation data here. Future work could investigate the sensitivity of the results to alternative options, including the use of process-informed (rather than purely statistical) methods currently in development.

## 3    Methods

### 3.1    Hydrological model

The Grid-to-Grid (G2G) is an area-wide runoff-production and routing model which typically uses a 1km grid and 15-minute timestep, and employs digital datasets to simulate the natural flow response to rainfall across the model domain, producing river flow outputs on a 1km grid aligned with the GB national grid (Bell et al., 2009). G2G uses 1km flow networks (Davies and Bell 2008), derived from the UK 50m Integrated Hydrological Digital Terrain Model (Morris and Flavin, 1990) using the network-derivation scheme of Paz et al. (2006). The routing component of G2G applied here uses kinematic wave approximations for sub-surface land and river flow paths and for surface land flow paths, but the Horton–Izzard nonlinear storage approach for surface river flow paths (Bell et al., 2009). In urban and suburban areas, identified through the LCM2000 spatial dataset of land-cover (Fuller et al., 2002), responsiveness is increased through the use of an enhanced routing speed and reduced soil storage, leading to a faster response to rainfall. While flows are simulated for every 1km cell, they are only output here for cells with a catchment drainage area of at least 50km$^2$, due to the use of daily driving precipitation data.

G2G has been widely tested and applied to explore climate change impacts on river flows across GB, for both floods (Bell et al., 2009, 2012; Lane and Kay 2021; Kay 2022; Kay et al. 2023) and droughts (Kay et al. 2018; Rudd et al., 2017, 2019; Lane and Kay, 2021). G2G is used for operational flood forecasting by the Flood Forecasting Centre for England and Wales (Price et al., 2012) and by the Scottish Flood Forecasting Service for Scotland (Cranston et al. 2012), and a substantial amount of model evaluation has been performed to support operational use (e.g. Moore et al. 2006, 2012, Cole et al. 2013, Wells et al. 2016).

Previous work assessing the performance of G2G driven by daily observed rainfall data across GB has shown generally good performance for median flows (Q50) and high flow volumes (Q5-Q30), but a tendency to underestimate flood peaks to the north/west with a more mixed picture, including overestimates, in the south/east (Kay 2022). Similar patterns were shown for simulation of the index flood (Formetta et al. 2018), where it was noted that "overestimation in southern and eastern Britain can, for many groundwater-dominated catchments, be attributed to the effects of artificial abstractions which are not currently included in the G2G" and that a "significant factor contributing to the underestimation is the contribution of short-duration intense rainfall events to peak river flows". Use of hourly, rather than equally-disaggregated daily, precipitation data was recently shown to improve performance for flood peaks, but to make little difference to simulated future changes in peak flows (Kay and Brown 2023). Beylich et al. (2021) also showed that hydrological modelling at an hourly time-step (with stochastic disaggregation of daily data) gave future flood changes of similar magnitude to modelling at a daily time-step, despite

underestimation of absolute values of flood peaks by daily modelling, for six catchments in central Germany (areas 39.1–823.5km$^2$). Only daily RCM precipitation data are available, so are used here, thus some underestimation of absolute values of peak flows is expected, but future changes in peak flows are assumed to be representative. The effect of any bias in RCM precipitation will also be moderated (to some extent) by the derivation of peak flow thresholds for each ensemble member separately (Section 3.2).

This application of G2G used gridded driving data provided by i) observations, and ii) the UKCP18 Regional Projections (Section 2), and provides gridded outputs of daily mean river flows. The results from the observation-based model run (SIMOBS) are used to enable an evaluation of the performance of the baseline climate model data for simulating the characteristics of widespread flood events. The G2G outputs of the SIMOBS run shows good agreement with observed flows (from NRFA) for extreme events in 1043 catchments across GB (Figure 1 shows the 20-year event based on the 1980-2010 time-slice). Bias varies spatially across Great Britain (Supplementary Figure 1) but mostly with positive bias in central areas.

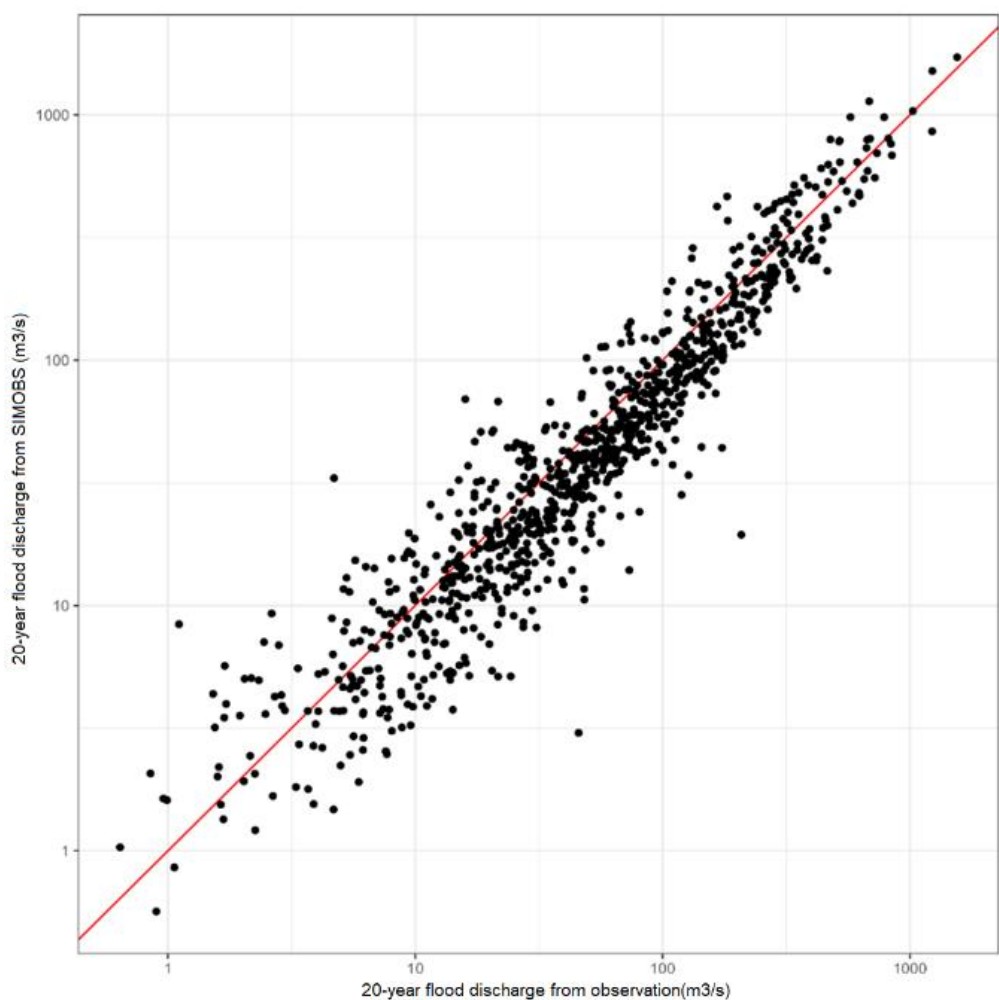

**Figure 1 Scatterplot of estimate of 20-year flood (stationary estimate based on 1980-2010) for GB gauging stations based on observations and observation-driven G2G simulation.**

**3.2 Event extraction**

For each RCM ensemble member, two time-slices were considered: 1980-2010 and 2050-2080, to serve as baseline and future viewpoints. Event time series were extracted using a peak-over-threshold (POT) approach as used by the NRFA (UK National River Flow Archive; Robson and Reed, 1999). In this approach, peaks are identified as exceedances above some predetermined threshold. To improve the independence of events, they must be sufficiently far apart (based on the average time-to-peak of 145 storm hydrographs). Additionally, consecutive events are checked to see if the minimum flow between the two peaks is less than two-thirds of both peaks, otherwise the lower peak is discarded (this process is iterated until no more events are removed).

To determine the most appropriate exceedance threshold to use at each 1km grid-square, five different percentiles of flow were investigated, ranging from five events per year to one event every 10 years on average. As a result, for each grid-square the following numbers of days were selected for each 30-year time-slice:

- 5 events per year (POT5)        – 150 days per grid-square
- 2 events per year (POT2)        – 60 days per grid-square
- 1 event per year (POT1)         – 30 days per grid-square
- 1 event in 5 years (POT0.2)     – 6 days per grid-square
- 1 event per decade (POT0.1)    – 3 days per grid-square

Note this is independent of the distribution of the data due to the use of empirical percentiles rather than fixed, absolute values of flow.

We define widespread events as timepoints for which a large number of locations experience very high flow (i.e. above the POT threshold) simultaneously. To determine when widespread events occurred, different levels of extent above threshold

were investigated to ensure that a good range of widespread events were captured, whilst ensuring that only events that could be described as "extreme" in some way were retained. To this end, the extent of an event was measured by the percentage of grid-squares on the river network which were simultaneously above their respective threshold values (denoted "inundated"). Five minimum extents were investigated: 5%, 2%, 1%, 0.5%, and 0.1%. Note that for the GB river network, 19,914 1km*1km grid-squares were considered as being in the network, so an extent of 1% corresponds to ~200 km$^2$ of inundated grid-squares.

To select the at-site threshold and minimum extent, all the combinations above were trialled on a single ensemble member (01) for the 1980-2010 time-slice, and the number of days fulfilling both inundation criteria (at-site threshold and minimum extent) over the 30-year time-slice are shown in Table 1. Very similar patterns of events extracted (not different at a statistically significant level) were observed for all of the ensemble members.

**Table 1 Number of days where national inundation according to a given threshold (rows) exceeds a certain percentage (columns). PoE = Daily Probability of exceedance**

|  |  | Extent lower threshold | | | | |
| --- | --- | --- | --- | --- | --- | --- |
| # exceedences | Daily PoE | 5% | 2% | 1% | 0.5% | 0.1% |
| POT5 | 5/360 | 839 | 1510 | 1981 | 2418 | 3427 |
| POT2 | 2/360 | 353 | 727 | 1027 | 1340 | 2031 |
| POT1 | 1/360 | 160 | 401 | 589 | 826 | 1345 |
| POT0.2 | 1/1800 | 25 | 77 | 144 | 239 | 444 |
| POT0.1 | 1/3600 | 14 | 35 | 74 | 117 | 262 |

The POT2 threshold and the 0.1% minimum extent were selected for the following reasons. POT2 provided a good balance between having enough exceedances to derive widespread events and keeping the threshold high enough to reasonably model

the peaks-over-threshold using an extreme-value distribution. The 0.1% inundation coverage was selected to ensure that small, very extreme events were not excluded. For applications in risk estimation, these events of smaller extent may occur in areas with high potential economic losses, and so are important for accurately estimating national annual damages.

     With this set of parameters for inundation, the specific at-site thresholds were calculated for each grid-square under each RCM ensemble member, using the thresholds from the 1980-2010 time-slice for both baseline and future events. This was to allow

the future events to be described in terms of baseline return periods. For the SIMOBS run, there is only a 1980-2010 time-slice, but the same approach was used to determine at-site thresholds.

     At this point, the event set consisted only of single-day events, which may not truly represent widespread events in the temporal sense, owing to the way in which storms move across a region over time and the typical time taken for water to travel downstream. To correct this, multi-day events were also defined. For the selected inundation threshold (POT2), event lengths

were defined as the number of consecutive days for which the extent exceeded the selected spatial limit (0.1%).

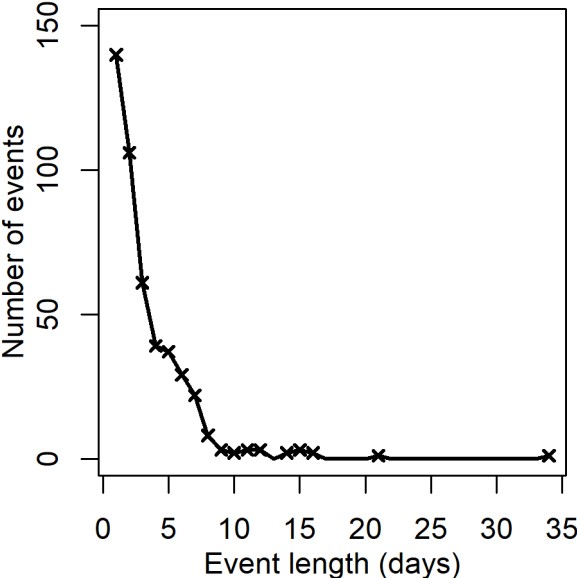

**Figure 2 Number of events with different durations, based on 0.1% extent lower threshold and at-site exceedance of two days per year (POT2).**

     For the RCM 01 ensemble member in the 1980-2010 time-slice, the distribution of event lengths is shown in Figure 2. Here it

can be seen that beyond seven days, there are very few events which fall under the definition above. There are some arguments that one should consider events up to 21 days (De Luca et al., 2017) but this may lead to a greater likelihood of two independent events of small geographical spread being recorded as a single, larger event. Such pairs (or larger groupings) of events may arise from different weather systems in, for example, the North-West and South-East of England. As a compromise therefore, events were limited to 14 days. If an event exceeded this time limit, the 14 days surrounding the day at which spatial spread

was highest were retained as "the event" (six before, seven after).

To keep the events simple to interpret, multi-day events were summarised. For each grid-square, each event was summarised by the highest single-day value at that grid-square during that event. Taken nationally, this retains the maximum flow at each point over the whole region, which should capture the most extreme flows within an event, and will also be helpful for estimating upper bounds of risk associated with such events. A more in-depth investigation into multi-day events could be the focus of future work.

To assess the change in spatial extremal datasets, one can investigate whether the spatial dependence changes between time-slices; $\chi$ and $\bar{\chi}$, two measure of extremal dependance (Coles, 2001), are calculated between pairs of points. $\bar{\chi}$ describes the level of asymptotic independance between two random variables if both are above given thresholds. $\chi$ complements this: if two random variables are asymptotically dependent, this describes the strength of that asymptotic dependence. For two points $i$ and $j$,

$$\chi_{i,j} = \lim_{x \to \infty} P[Q_i > x | Q_j > x]$$

If $C^*(u, v) = 1 - u - v + C(u, v)$, for a copula $C$, then

$$\bar{\chi} = \lim_{u \to 1} \frac{2 \log (1 - u)}{\log C^*(u, u)}$$

$\chi$ describes the level of asymptotic dependence; if $\chi > 0$ then the variables are asymptotically dependent, and $\bar{\chi} = 1$ automatically. But if $\chi = 0$, they are asymptotically independent. In this case, $\bar{\chi}$ describes the dependence for large but not asymptotic values of flow. $\bar{\chi}$ close to 1 indicates the variables are highly dependent except at the asymptotic limit.

## 3.3    Return Periods

To ensure a good fit of return periods for the most extreme events, the top 60 independent peaks in each ensemble member and timeslice were found using the peak-extraction algorithm as described in Section 3.2. For values over the threshold, a Generalised Pareto distribution (GPa) was used with distribution function

$$F_{GPA}(x) = 1 - \left(1 + \frac{\kappa(x - u)}{\alpha}\right)^{\frac{1}{\kappa}} = P[Flow > x | Flow > u]$$

with threshold $u$, scale parameter $\alpha > 0$ and shape parameter $-1 \leq \kappa \leq 1$. This was fitted to the series of independent peaks over the threshold to give a modelled daily probability of exceedance. u is the flow threshold at a specific location, and $P[x > u] = 2/360$ , since this investigation uses the POT2 threshold defined in Section 3.2.

To convert from a per-exceedance PoE to a more widely-used annual PoE, a simple scaling factor was applied based on there being 60 events per location over 30 years:

$$p_{ANNUAL} = p_{EVENT} \times \left(\frac{60}{30}\right)$$

In the rest of this work, plots are presented using annual probabilities of exceedance. Due to the limits of using 30-year time-slices of data, return periods are capped at 1000 years since the uncertainty on exceedance probabilities is very high for the most infrequent events.

## 4    Results

Figure 3 shows four example events extracted from the 1980-2010 time-slice from RCM ensemble member 01, with return periods described in years. The coloured extent of an event was restricted to those points with a daily probability of exceedance of less than 2/360. On the whole, the events are spatially contiguous, and the example events suggest that return period is
highly peaked around one location and quickly tapers off away from the "epicentre". These are four of the largest events in the 1980-2010 time-slice and show a broad range of different events covering Scotland, southern England and central England, with key patches of very extreme flow in Figure 3b-d, whereas **Error! Reference source not found.**a shows a widespread but less severe event (in terms of return period of flow). In the rest of this section, return periods reported in the text and figures are the maximum return period observed (across space and time) within a single event. The analysis focuses on the differences
between past and future and across space, though differences between the RCM ensemble members should be mentioned.

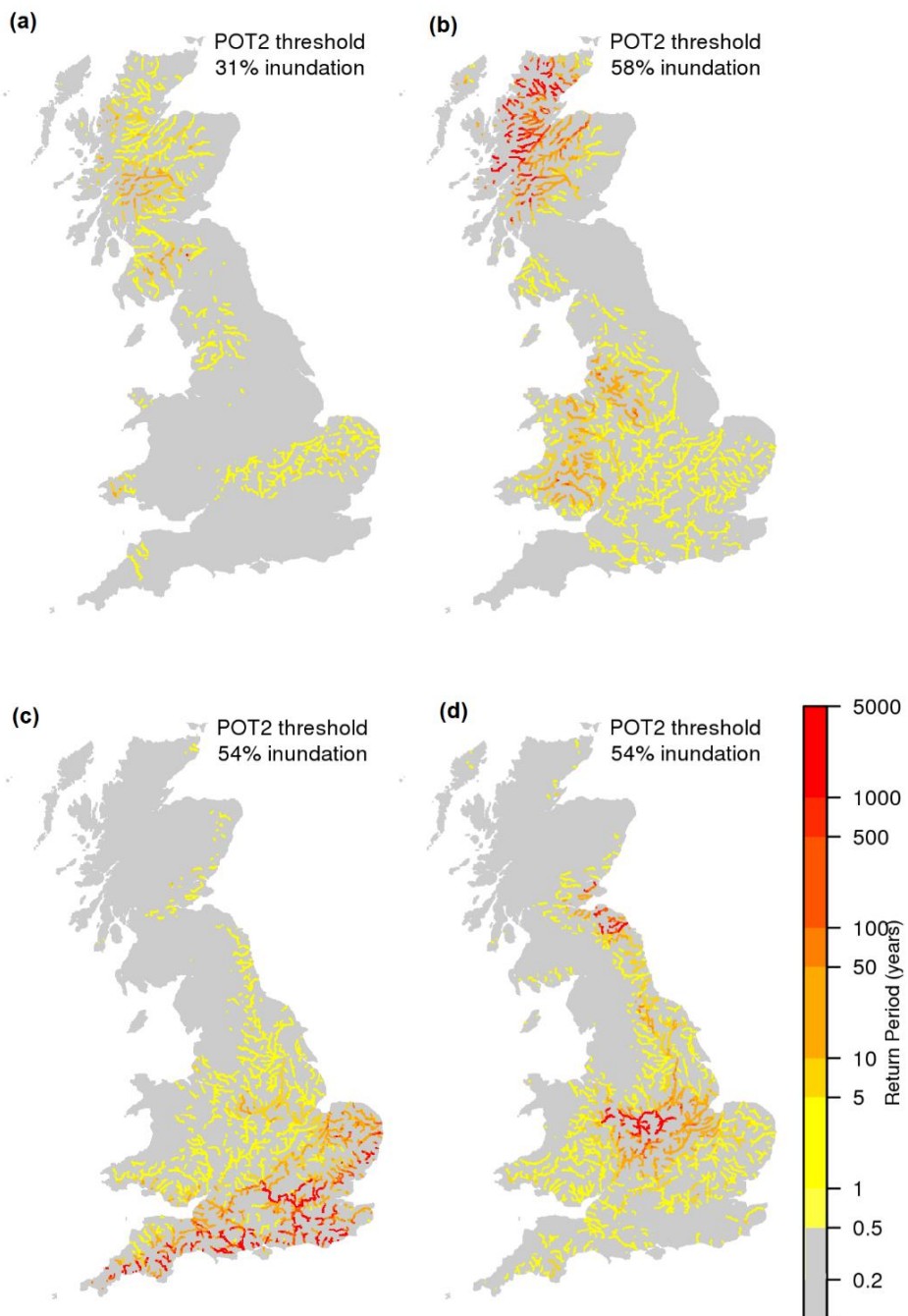

**Figure 3 Example events from 1980-2010 time-slice from a single RCM ensemble member, showing return period in years. The percentages shown refer to the number of river grid cells 'inundated' (i.e. the percentage of river grid cells where flow exceeds the POT2 threshold), not the percentage of GB land area flooded.**

Figure 4 shows that the event areas are fairly consistent between the RCM-driven runs and the SIMOBS run, although with a bias in the baseline RCM-driven runs to larger events with lower return periods. The RCM-driven runs show a slightly flatter distribution of return periods in the 2050-2080 time-slice, with little change in the distribution of areas. Supplementary Figure 2 shows how the results vary between ensemble members. Ensemble members 07 and 08 show a slightly more uniform distribution of events across log(Area), and ensemble member 11 shows a slightly higher number of small events, around 200 events with a footprint of less than 100km$^2$. Ensemble member 01 shows the greatest difference between the 1980-2010 and 2050-2080 time-slices (more than 50 fewer events with return period less than 8 years).

In the rest of this section, the event sets from all ensemble members are combined and given equal weighting. In the supplementary material, ensemble members are treated as separate sources of equal weighting. Ensemble members are shown in different colours and have the convex hull of the points from each ensemble member highlighted to show in particular variation in the extremes.

Taking the union of events extracted from all the ensemble members, changes in extent and duration can be examined. Figure 5 shows the the number of widespread events by boreal season. There is relatively similar seasonal pattern between SIMOBS and the RCM baseline (with the RCM ensemble range encompassing the SIMOBS numbers, except in Autumn), although the RCM tends to accentuate the seasonal pattern of more flood events in winter and fewer in summer. Overall there are a greater number of widespread events in the future (7553) than in the baseline time-slice (7225 events). However, in the months of March to August, and particularly in June to August (boreal summer), there are fewer widespread events in the future time-slice.

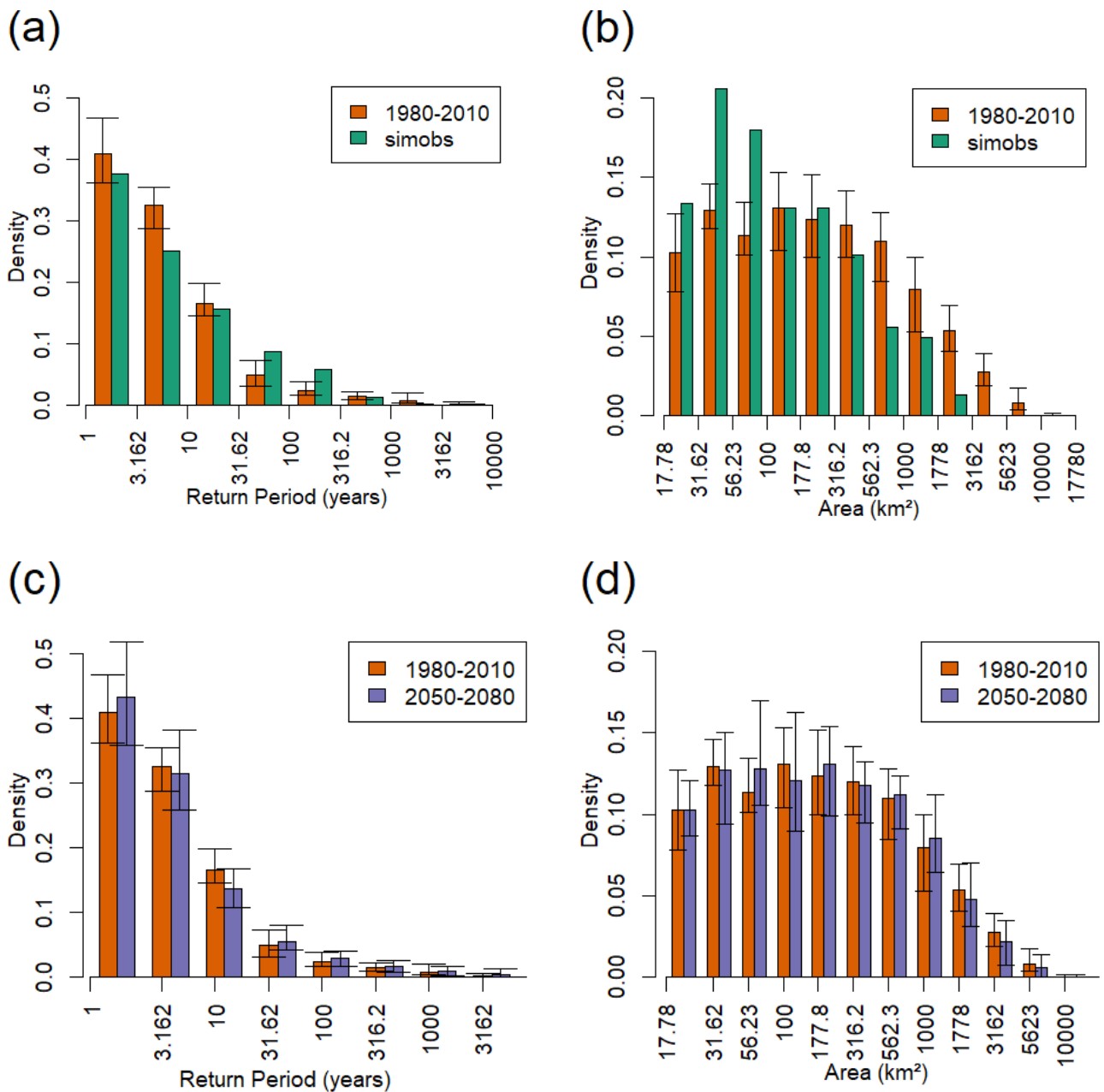

Figure 4 Barplots comparing the distributions of a) event return period and b) event area for the SIMOBS run and the baseline time-slice of the RCM-driven runs (averaged across ensemble members), then c) event return period and d) event area for the baseline and future time-slices of the RCM-driven runs (averaged across ensemble members) . The error bars show minima and maxima across RCM ensemble members for baseline and future time-slices. Note SIMOBS has only one run.

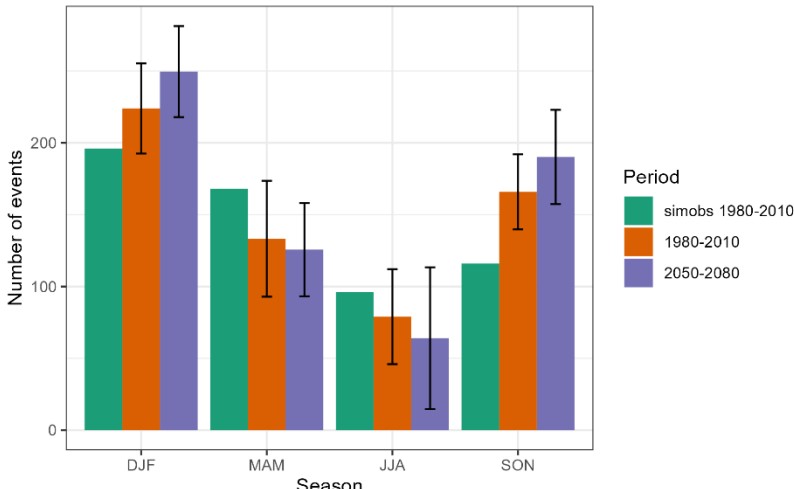

**Figure 5 Barplots comparing the number of widespread events, split by season, for the SIMOBS run and the baseline and future time-slices of the RCM-driven runs (averaged across ensemble members). The error bars show minima and maxima across RCM ensemble members for baseline and future time-slices.**

In terms of event duration, Figure 6 shows how this varies by season and time-slice, and how that is linked to return period. The SIMOBS run appears to generate shorter events on average compared to the RCM-driven runs, suggesting a slightly

stronger temporal autocorrelation in the effects of the use of UKCP18 input data. The return periods (as seen in Figure 4) are broadly similar in distribution. The figure suggests that duration and return period are somewhat correlated, in that the longest duration events are very unlikely to have a low return period (i.e. to occur frequently). However, there are a number of events which are of short duration but high return period. As one might expect, events are shorter in summer (JJA), with very few summer events extending longer than 5 days. In the future time-slice, event duration seems to be slightly shorter on average,

and this is more pronounced in spring (MAM) and summer (JJA) , reducing from 3.54 to 2.99 days in spring, 2.20 to 2.04 in summer. The events with the highest return periods are in autumn (SON) and winter (DJF), in both time-slices, though the distribution of return periods in the future has heavier tails (note the return period axis is on a logarithmic scale). Supplementary Figure 3 shows that there is some variability between ensemble members, particularly in the extremes, but the overall pattern is preserved throughout, as expected from Figure 4.

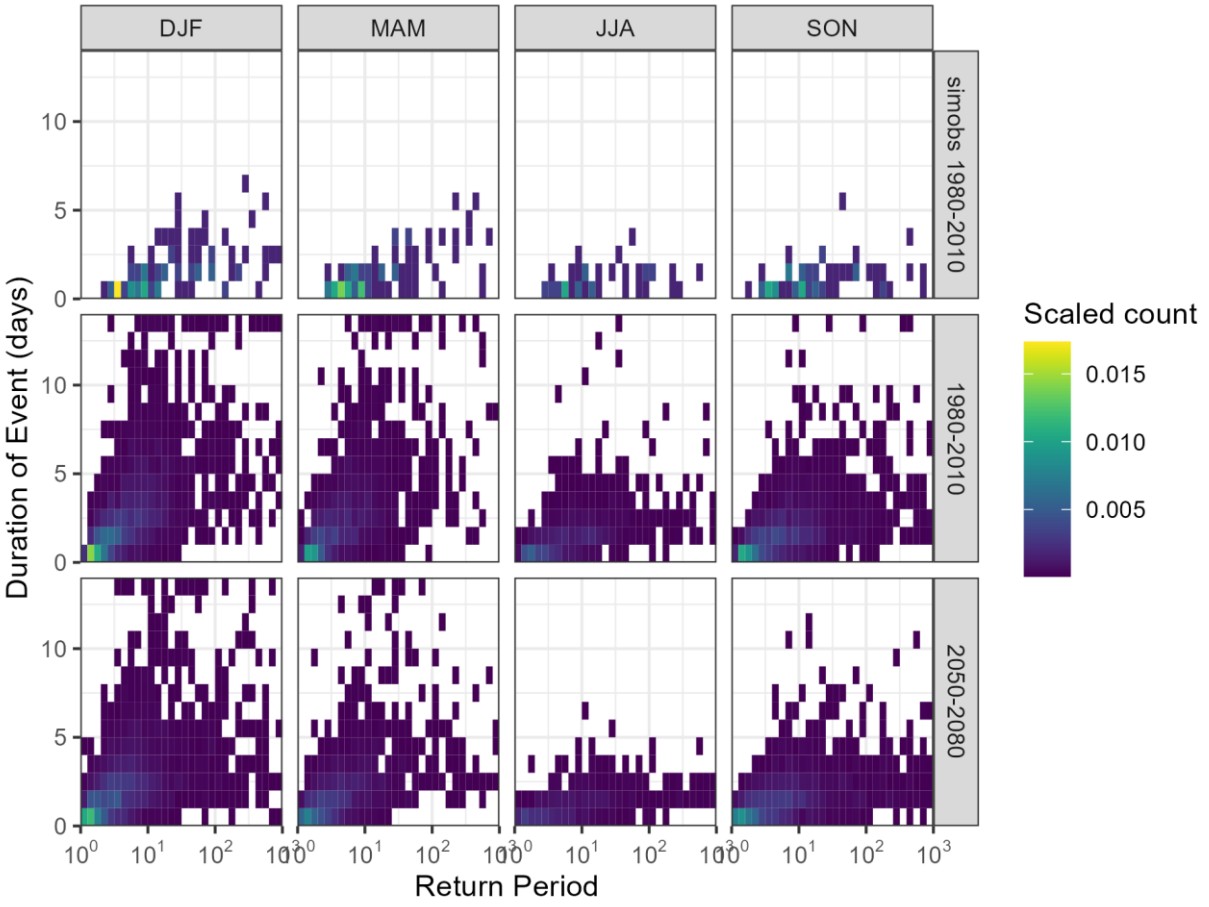

**Figure 6 Heatmaps showing joint distribution of return period and event duration, split by season, for the SIMOBS run and the baseline and future time-slices of the RCM-driven runs (summed across ensemble members). Count is scaled so that the total of events in each time-slice adds up to 1.**

Figure 7 shows how area and peak return period vary by season in the two time-slices, and in comparison to the SIMOBS run.

As one might expect, there is a correlation between area and peak pointwise return period across both RCM-driven time-slices. The changes between the two time-slices are subtle, but there is an overall trend towards an increase in the range of peak return period: the 95[th] percentile of return periods increases in all seasons, from an increase of 10 years in spring to 205 in summer, with the 5[th] percentile being ~1.2 in all seasons and timeslices. The extent of widespread events appears to stay consistent between the 1980-2010 and 2050-2080 time-slices, with a possible slight reduction in extent of the largest events in the future

summer. In all seasons there are a small number of events with return periods exceeding 1000 years, particularly in winter and autumn. Supplementary Figure 3 shows that this pattern is matched between ensemble members, but there is some variability in the relative patterns of duration and rarity in the extremes.The SIMOBS run shows a broadly similar distribution to the baseline (1980-2010) timeslice, although the variability and reduced smoothness appears to be increased, due to the much smaller number of events from that single run (~500 compared to ~7000 from all 12 RCM ensemble members).

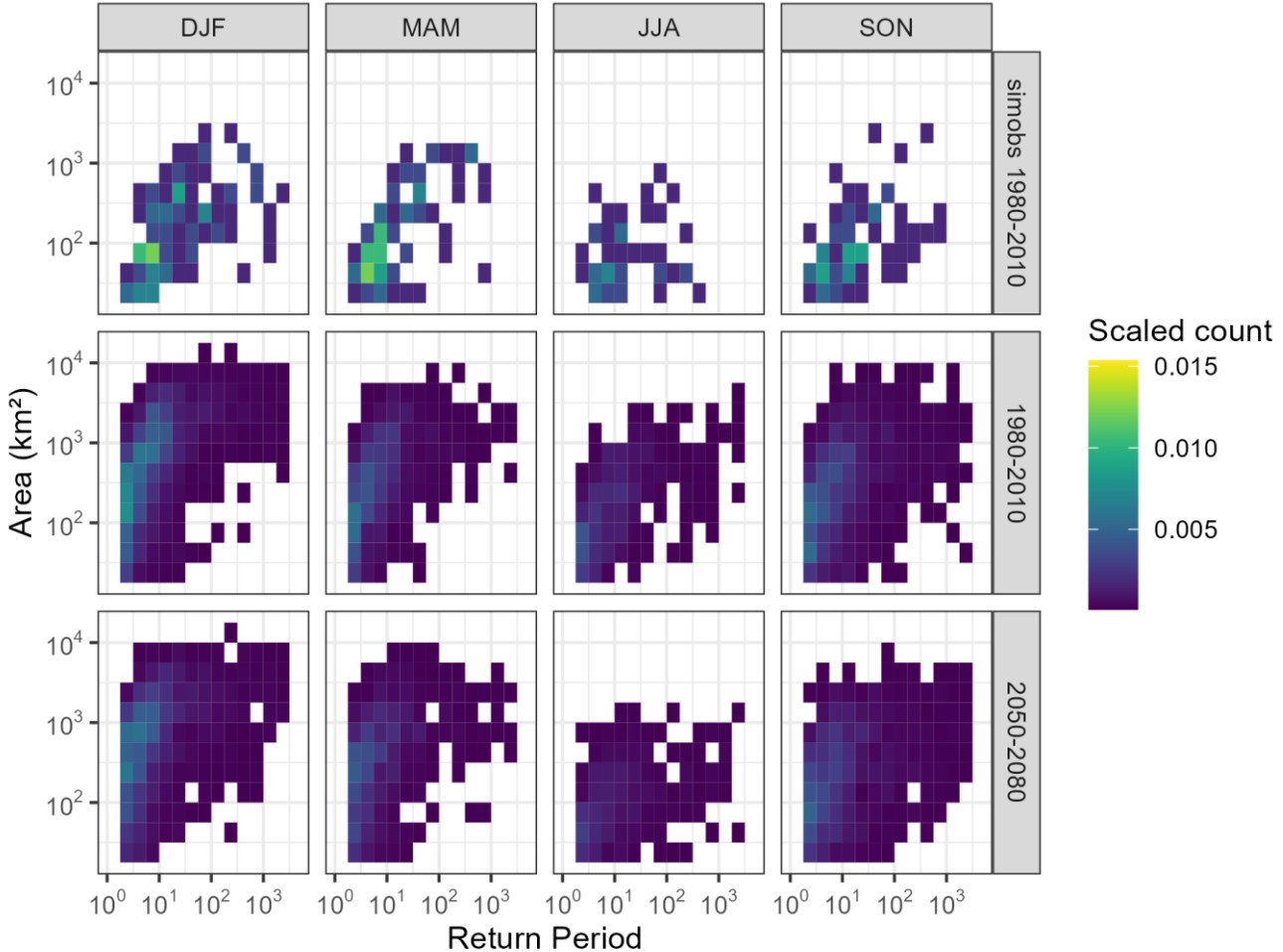

**Figure 7 Heatmaps showing distribution of events with different areas and return periods, , split by season, for the SIMOBS run and the baseline and future time-slices of the RCM-driven runs (summed across all ensemble members). Count is scaled so that the total of events in each time-slice adds up to 1.**

Figure 8 shows how dependence varies between pairs of points across the river network. Here asymptotic dependence appears to have a limit at most location pairs of around 120km ($\chi$ is only shown for pairs of locations for which the upper bound of a bootstrapped uncertainty bound of $\bar{\chi}$ exceeds 0.99). The figure suggests that asymptotic dependence decreases as distance increases. In the asymptotically independent case (Figure 8 right), we see a similar pattern in dependence for large values of flow, with high dependence at short distances, even if they are independent in the limit. There seems to be little change in dependence between the two time-slices, although the asymptotic dependence appears to extend slightly further in the baseline time-slice. When compared with the SIMOBS run, a shorter radius of asymptotic dependence is exhibited, and at a distance of around 200km, values of $\bar{\chi}$ are more concentrated around 0.5 compared to the baseline and future time-slices.

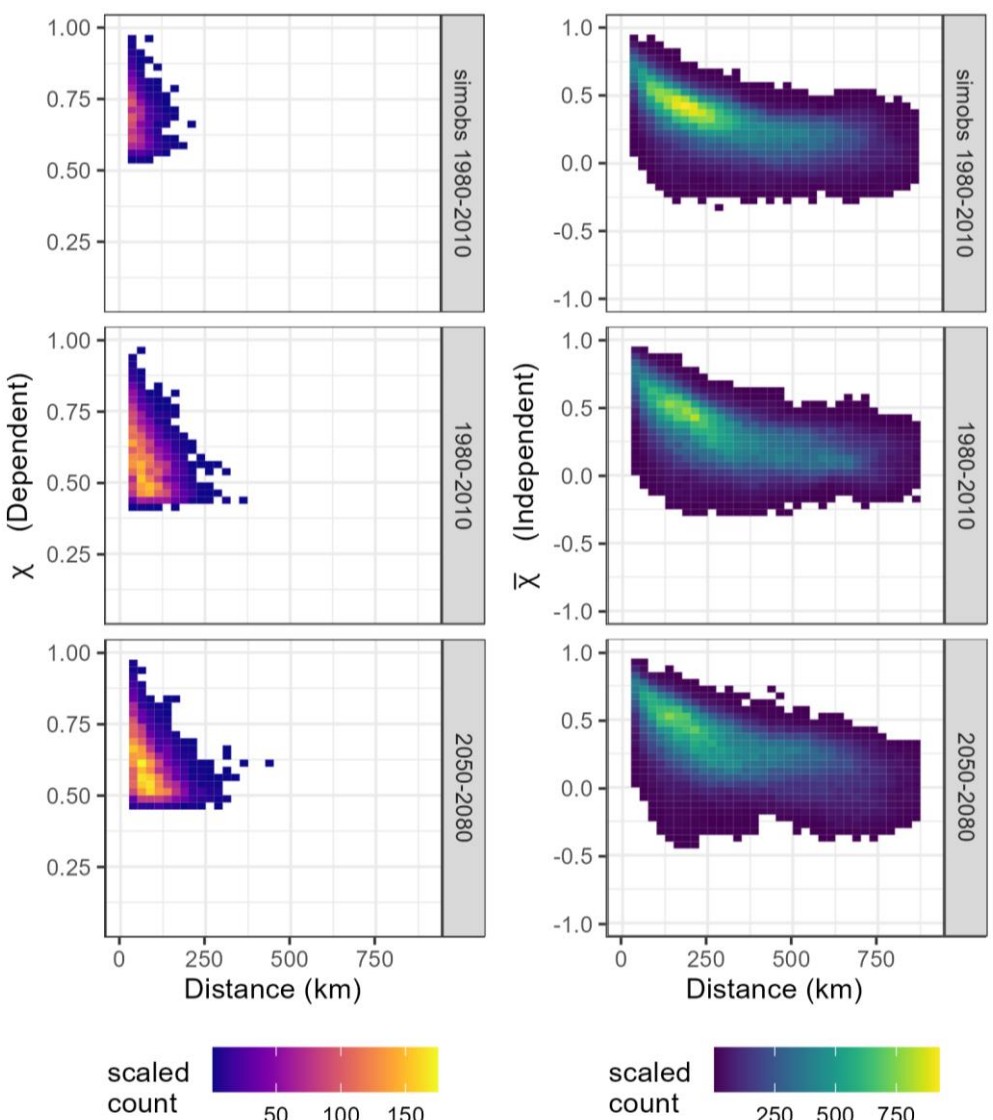

**Figure 8 Heatmaps showing asymptotic dependence for 100 000 random pairs of points on the river network. $\chi$ is only shown for pairs of locations which are asymptotically dependent based on $\bar{\chi} > 0.99$.**

If the events are subdivided by season, subtle differences can be observed (Figure 9). Overall, spring and summer show less asymptotic dependence (lower values of $\chi$ and $\bar{\chi}$) than autumn and winter: mean $\chi$ is 0.641 and mean $\bar{\chi}$ is 0.222 for spring and summer (Mar-Aug),, compared to mean $\chi$ 0.673 and mean $\bar{\chi}$ 0.363 for autumn and winter (Sep-Feb). Also, the 50% contour for $\bar{\chi}$ is longer in spring (MAM) (max distance of 495km in baseline, 545km in future) than summer (JJA) (max distance of 431km in baseline, 462km in future) in both time-slices, suggesting that the variance in $\bar{\chi}$ exhibits seasonal variation. Between baseline and future, as for Figure 8, the differences are marginal, but both $\chi$ and $\bar{\chi}$ show smaller 50% contours in autumn compared to the other seasons, suggesting reducing variation in asymptotic dependence in this season. For other percentile

contours, patterns are very similar and follow the shapes of Figure 9. For the SIMOBS run, the smaller sample size (one run compared to multiple ensemble members) leads to a less regular contour. Overall, within- and between-seasonal variability is

higher in the SIMOBS runThis is also mirrored in Supplementary Figure 5, which shows this split by ensemble member, where spatial variation in coherence is strongly preserved between ensemble members.

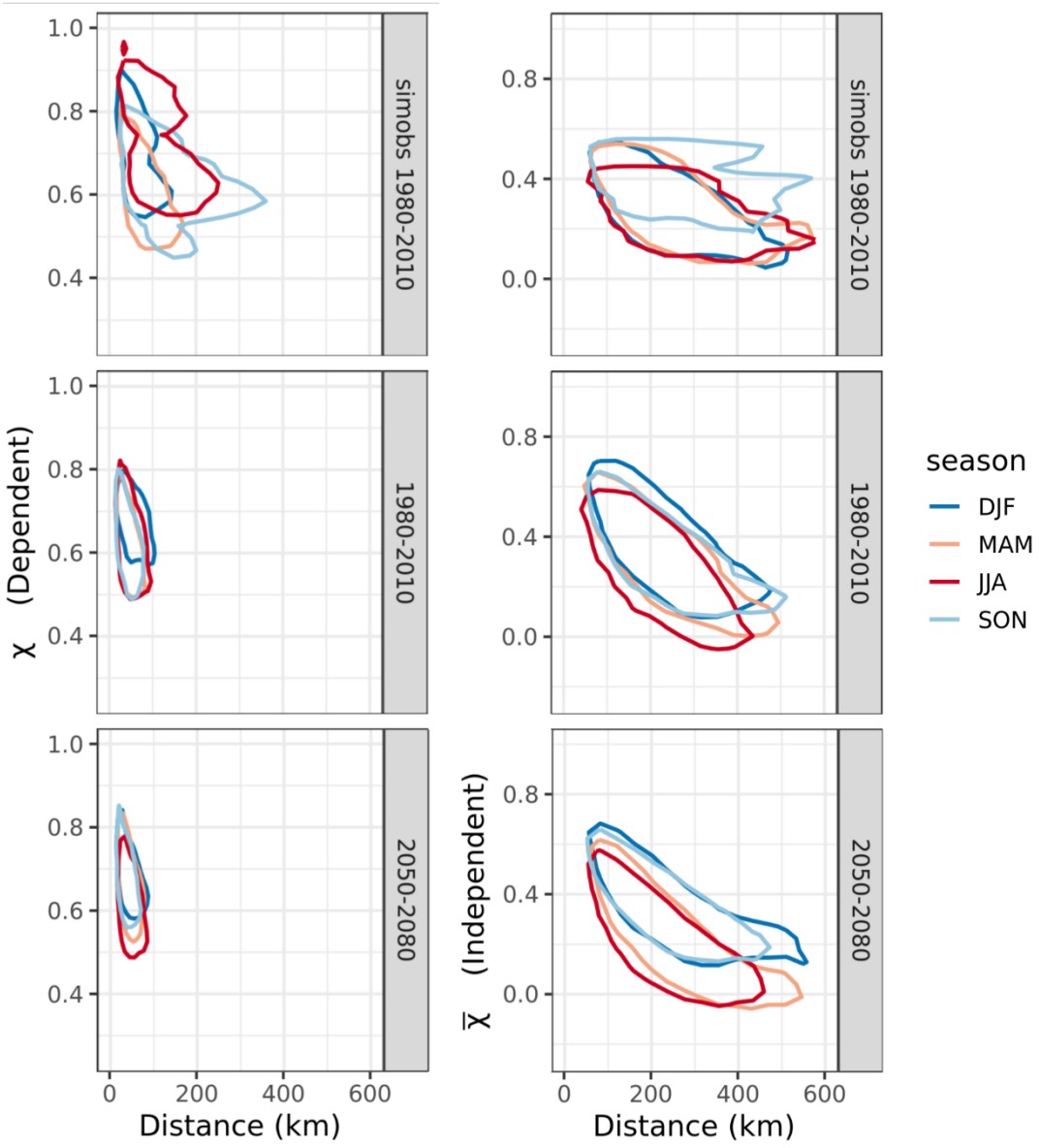

**Figure 9 Contour showing asymptotic dependence for 100 000 random pairs of points on the river network. Contours show smallest area that contains 50% of point-pairs, split by season and time-slice. $\chi$ and $\bar{\chi}$ as in Figure 8**

## 5    Discussion

### 5.1    GB results

A non-trivial number of very extreme events were observed in the RCM-driven runs for both baseline and future time-slices. Tawn et al. (2018) point out that, within the observed annual maxima series, the chance of a 100-year return period flood event occuring somewhere within a set of 916 gauging stations in England and Wales is approximately 78% in any one year, and so over a gridded dataset of more points, and with more years, the occurrence of these rare events is statistically plausible. Also, due to the probability distributions used, a small change in event peak flow magnitude in the upper tail of the distribution can lead to a large change in return period (when the shape parameter κ is positive, which is the case for most of the UK (Griffin et al., 2019)). Variability in extent, duration and return period in the modelled baseline and future time-slices seems to be much increased compared to the SIMOBS runs, but this may be in part due to the greatly reduced sample size in the SIMOBS run (1 run vs 12 ensemble members). The present and future time-slices do seem to overestimate duration and rarity compared to the SIMOBS run, which may be due to a difficulty in capturing smaller, less extreme events in the UKCP18 data and the subsequent model outputs.

The number of widespread events (based on a POT2 threshold derived from 1980-2010 data) was found to increase in total in the future time-slice (Figure 5) but was slightly lower in the future for spring and summer (March-August) events. The typical spatial extent of events was found to be fairly consistent between time-slices, but summer (June-August) events appeared smaller in the future across all return periods. A pairwise analysis suggested that inundated locations were asymptotically independent beyond a radius of around 120 km, but the distribution of dependence was slightly less concentrated in the future. Event duration decreased on average in all seasons between the two time-slices. Patterns were similar across all RCM ensemble members. This suggests an increase in seasonality in widespread flood events, with more widespread flooding in winter, and possibly a shift towards smaller intense flooding in spring and summer.

The increased number of widespread flood events is consistent with work by Lavers and Villarini (2013) which shows the possible increase in atmospheric rivers, especially in Western Europe, which drive extreme precipitation events. The limited change in extent of widespread events in future is consistent with Bevaqua et al. (2021), who analyse multi-thousand-year climate model simulations and show non-significant changes in the spatial footprint of winter extreme precipitation over most of Britain, although there are significant changes across most of the rest of the Northern Hemisphere extratropics. The seasonality changes are consistent with Kay et al. (2022), who show that projections of soil moisture change point towards drier summers and autumns, and Blöschl et al (2017), who suggest that UK floods are closely linked to soil moisture timing. But seasonality changes may also be linked to a change in the size of flooding events in the future in summer which, in the UK, are typically linked to short-duration, intense summer storms. It may be that these intense storms become smaller in extent, below this paper's definition of "widespread", but Chen et al. (2021) suggests that convective storms may cover a greater area in the future. Between ensemble members (Supplementary Figure 4), variability of flood extent is higher in summer (JJA),

even more so in the future time-slice, and one member actually shows an increase in summer (JJA) events in the future time-slice (Figure 5).

The simulated future changes in the characteristics of widespread flood events have to be interpreted in the context of some differences in event characteristics for the baseline climate projection-driven model runs compared to the observation-driven model run. The baseline RCM-driven runs tend to give larger events with lower return periods, and accentuate the seasonal pattern of more events in winter and fewer in summer. Differences could be due to RCM biases, but are also probably related to the 12km resolution of the RCM data, which likely means that rainfall inputs occur more homogeneously and simultaneously over larger areas than for 1km observed data, leading to flood events of typically larger extent but lower return period. Future work could investigate the sensitivity of the results to bias correction options, including use of process-informed (rather than purely statistical) methods currently being developed. The UKCP18 Local data (Kendon et al. 2021) could also be applied; this provides an ensemble from a higher resolution (2.2km) convection-permitting model (CPM), nested in the RCM ensemble, and has been shown to give some differences in at-site changes in flood peaks (Kay 2022). It would be interesting to assess whether there are also differences in seasonal widespread flooding characteristics, both for the baseline time-slice and in terms of possible future changes.

## 5.2    Methodology

While the focus here was GB, the methods and analyses described could be applied to other regions with hydrological models and climate projections of appropriate resolution. The most suitable definition for a "widespread flood event" may vary for other countries/regions. For example, other applications may only be interested in events with more extreme flows, so the POT threshold could be increased, or only interested in events with much greater extents, so the extent threshold could be increased. The event duration would also need to be considered in the context of basin size, typical event flashiness (related to, for example, geology and steepness) and typical storm duration for the region.

The event extraction method developed here did not explicitly require events to be spatially contiguous. While this does generally appear to be the case for the largest events, the event shown in Figure 3a suggests that some potentially simultaneous but independent events are captured. This may be due to capturing two consecutive or overlapping events, due to the method of event length determination, thus a more sophisticated form of event delineation could improve the process. Brunner et al. (2020) make use of a spatial dependence function (F-madogram) and hierarchical clustering to determine events for which points are mutually dependent to a sufficient degree, which could improve event identification.

In addition, future work could expand on the methods developed here by investigating ways of expanding the set of widespread flood events using computationally-cheaper methods such as emulation or statistical Monte Carlo methods.

To highlight spatial dependence in a simpler way than χ, Berghuijs et al. (2019) use a metric of flood "synchrony", measuring how often extreme floods occur at the same time within a given radius of a target point. Gridded datasets like those used here could be evaluated using this metric, or one like it.

# 6    Conclusions

This paper has used the latest regional climate projections for the UK (UKCP18 Regional) and a national-scale grid-based hydrological model (Grid-to-Grid, G2G) to generate grids of daily mean flows across GB, from which a set of widespread flood events was extracted. The question of what defines a "widespread flood event" was investigated; here these were defined as exceeding an at-site 99.5th percentile (equivalent to two days per year, POT2) simultaneously over an area of at least 20 km$^2$, with a maximum duration of 14 days. The event set was used to investigate potential changes in spatial structure of river

flooding across mainland GB between the 1990s and 2060s. In summary, the number of widespread events was found to increase in total in the future time-slice, but was slightly lower in the future for spring and summer (March-August) events. The typical spatial extent of events was found to be fairly consistent between time-slices, but summer (June-August) events appeared smaller in the future across all return periods.

While the focus here was GB, the methods and analyses described could be applied to other regions with hydrological models

and climate projections of appropriate resolution. The most suitable definition for a "widespread flood event" may vary for other countries/regions. The analysis of Bevaqua et al. (2021) suggests larger spatial footprints of winter precipitation extremes in future across much of the Northern Hemisphere (apart from Britain and parts of western Europe and Africa), which could lead to larger widespread flood extents in such regions. Future changes in widespread flooding could have important implications for flood incidence response and flood risk management, for example insurers want to be able to understand and

mitigate their risk of receiving concurrent claims from large numbers of homes and businesses. However, the potential future changes have to be interpreted in the context of any differences in event characteristics between the baseline climate projection-driven model runs and an observation-driven model run.

This paper and the data generated therein forms the basis for a wider scheme of work generating extreme flooding events for risk analysis, which is the subject of a number of subsequent papers: Griffin et al. (2022b) discussing statistical methods to

generate large numbers of plausible widespread events with long return periods and Sayers et al. (2023) on applying the statistical event sets to risk analysis through catastrophe modelling methods. Further work could look at more sophisticated methods of event identification, and look at describing or separating simultaneous or temporally-overlapping events. This work focuses on fluvial flooding but surface water flooding (not from inundation of rivers and water bodies) is also a large factor in estimating economic losses due to flooding. It would be of interest to use the Grid-to-Grid model including surface water

(Rudd et al., 2020) applied to the framework of this paper to see if the different types of flooding will change in different ways over time.

## Data availability

Peak flow data freely available from UK National River Flow Archive (nrfa.ceh.ac.uk). UKCP18 data available from Met Office under and Open Government Licence. Event set can be found at the Environmental Information Data Centre (Griffin et

al., 2022a).

**Author Contributions**

ES and PS managed the project, AK and VB ran the hydrological modelling, AG ran the event extraction and summary, and performed the statistical analysis. All authors assisted in writing and editing the manuscript.

**Acknowledgements**

Funding for the project was provided through the UK Climate Resilience Programme supported by UK Research and Innovation and the UK Met Office.

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
