# Peer review of "Widespread flooding dynamics under climate change: characterising floods using grid-based hydrological modelling and regional climate projections"

_Hydrology and Earth System Sciences, 2022_

## Author Comment (AC2)

Reviewer 2

- The authors present a study on flood changes under climate change in the UK. The focus lies specifically on changes in modelled flood return periods based on an ensemble climate projection. The main point of analysis is the changes in widespread flooding. The authors find that there is more widespread flooding in winter and less in summer in the future projected climate. Further analysis included changes in return period, area covered and duration of events between current and future climate.
- Overall, I like that the article focuses specifically on simultaneously occurring flood events under climate change. The analysis and results presented here show thorough, good work. I would have wished for a bit more focus on how the uncertainty of the climate ensemble translates into the results and more discussion of the results regarding potential drivers of change. While I have lots of comments and open questions, all of them are minor and should be quick to address.

Introduction

- Since a large part of your results section talks about spatial dependence, can you motivate this analysis in the introduction? Especially since several people have already written about flood coherence/synchrony (Brunner et al. 2020) including results for the UK (Berghuijs et al. 2019).
  - On line 39 we add the following: "… more frequent and severe. Spatial coherence of flooding events – whether flood timings at different locations have become more correlated – is of key interest to national-scale actions to mitigate the associated loss. The dependence structure of river flow has been analysed on a Europe-wide scape (Berghuijs et al., 2019) and focusing on the United States (Brunner et al., 2020), focusing on synchrony of events within a given range."

Methodology:

- Can you elaborate on why you chose the Grid-to-Grid model for this analysis and how well it performs in streamflow/flood prediction under the current climate? This would allow some estimate how reliable future projections might be.
  - We will add some further information to Section 3.1 about previous studies using Grid-to-Grid, which has been widely tested and applied to explore climate change impacts on river flows across GB, including for floods and droughts (Bell et al., 2009, 2012; Kay et al., 2018; Rudd et al., 2019, Kay 2021, Lane & Kay 2021), and which is also used by the Environment Agency for flood forecasting in England (Price et al., 2012).
- I like that you thoroughly elaborate on your choice of thresholds regarding POT and inundation extend. Can you supplement this with a sentence along the lines of "Widespread events are defined as…".
  - Agreed. We add the following at line 95: "We define widespread events as timepoints for which a large number of locations experience very high flow (i.e. above the POT threshold) simultaneously."
- Can you elaborate more on the method chosen for asymptotic dependence and, more importantly, elaborate on what that means? I have not encountered this method before, nor did I understand by the end of the paper, what it actually tells me. If you are interested in using an established method, I can refer you again to the papers by Brunner et al, (2020) or Berghuijs et al. (2019). Their results should also be discussed in line 254 since it relates to your proposed further work.

- The measures of asymptotic dependence are well established but often misunderstood. To alleviate this, we edit the text at line 136 to the following: "…are calculated between pairs of points. For two points i and j, $\chi_{ij} = \lim P[Q_i > x \mid Q_j > x]$ as $x \rightarrow \infty$. If $C^*(u,v) = 1 - u - v + C(u,v)$ for a copula between two points i and j, then $\chi^* = \lim 2 \log(1-u)/\log(C^*(u,u))$ as $u \rightarrow 1$. Chi describes the level of asymptotic dependence, if chi == 0 then the variables are asymptotically independent, otherwise they are asymptotically dependent. In the asymptotically independent case, chibar describes the dependence for large but not asymptotic values of flow. In the asymptotically dependent case, chibar == 1."

- To comment on both panels of Figure 7, we change the sentence on line 220 to: "The figure suggests that asymptotic dependence decreases as distance increases. In the asymptotically independent case (Figure 7b), we see a similar pattern in dependence for large values of flow, with high dependence at short distances, even if they are independent in the limit."

- On line 254, we add the following: "…event length determination. Brunner et al. (2020) make use of a spatial dependence function (F-madogram) and hierarchical clustering to determine events for which points are mutually dependent to a sufficient degree. This would be an interesting direction to go in to improve event identification. To highlight spatial dependence in a simpler way than $\chi$, Berghuijs et al. (2019) use a metric of flood "synchrony", measuring how often extreme floods occur at the same time within a given radius of a target point. The gridded data set we have available here could be evaluated using this metric, or one like it."

Results:

- There seems to be a mix between results and discussion in the results section (e.g. lines 184-190 are discussion, not results). You could either call the results section "Results and Discussion" or move any discussion from the results section to "Discussion and Conclusion". Generally, the discussion could be more elaborate (see below).
  - We will call it Results and Discussion.
- You quite often talk about an "increase in the range", "little change", "less asymptotic dependence", "extend slightly". Can you support these statements with numbers?
  - We will add percentage changes where appropriate throughout the results and the discussion, and add change in $\chi$ and $\chi$bar as well.
- Line 205: "On the right of some panels (future winter and autumn) is a set of events with a peak return period of at least 1000 years." From what I see, all panels have events up and over a return period of 1000 years.
  - We change this sentence to "We see that in all seasons are a small number of events with return periods exceeding 1000 years, particularly in winter and autumn."
- Even though you use a climate ensemble as input data for the hydrological model, the presented results mostly do not give an overview of the uncertainty the different climate projections introduce. Can you please give an indication of how the ensemble spread demonstrates uncertainty in the results? Especially since you state in the abstract: "Results were consistent across ensemble members, with none showing significant difference in distribution." Since the two main conclusions are about the seasonal shift and spatial dependence, the results in Figure 3 are not enough to support this statement across all findings.
  - Aside from Figure 3 which is already split by ensemble member, we can easily include uncertainty bounds on Figure 4, and include some measure of variance in Figures 5, 6 and 7 through adjusting transparency (alpha), where low variance is shown by a stronger colour, and high variance by fainter colours. Including upper and lower bounds in addition would result in a lot of extra figures, or much more complex ones, at a cost to readability. At line 177 we replace the sentence "In the rest of this section, the event sets …" is replaced by "In

- the rest of this section, ensemble members are treated as separate sources of equal weighting. Variance between ensemble members is indicated in figures by brightness, and the colour indicated the median value of the respective statistic amongst the ensemble members.
- Figure 2: Can you include in the caption what the percent inundation refers to? Is this percent grid cells or percent land area?
  - We add the following on line 160: "The percentages show refer to the percentage of the number of river grid cells, not a fraction of UK land area."
- Figure 3: I would prefer if you would present a summary figure for the different model ensembles. After all, since the ensemble runs represent uncertainty, only presenting, comparing and analysing individual ensemble members does not make sense.
  - As mentioned above, we can add uncertainty bounds to Figure 4, and add levels of variance to figure 5-7. However, Figure 3 is important to highlight that we have not erroneously included ensemble members which differ significantly from the others. The specific statistics are covered more broadly in the subsequent figures, so we feel adding to this information-heavy figure would reduce readability.
- Figure 4: Since you are using ensemble results, can you include uncertainty bars into the event count? Secondly, the caption says that you take the sum of all ensemble results. I would think that the mean or median (and potentially even the range) is the more appropriate measure. This is the case for Figures 5 and 6 as well.
  - See above for our response on this
- Figure 5+6: Is there a specific reason why you have return period once on the x-axis and once on the y-axis? If not, I would recommend choosing one or the other, not both.
  - This is a formatting choice. In Figure 5, duration is much narrower in range than return period, so the heat maps have a "vertically" stretched profile. Conversely, the key features in figure 6 vary more by return period, and so again, due to the number of panels, the order was chosen to keep the figures clear.

Discussion:

- Although the analysis itself does not focus on drivers of change, there have been several published articles on how hydrology and specifically floods are changing in the UK. I think the discussion would benefit from discussing the results of this study in the context of previous findings. For example, there is a projected increase in winter atmospheric rivers in the UK which are likely to bring widespread flooding (Lavers et al, 2013). Furthermore, floods in the UK are strongly associated with soil moisture timing (Blöschl et al, 2017). Do changes in the soil moisture influence in the increase/decrease of widespread flooding in the UK?
  - We will add a couple of sentences to the discussion (at line 244) to provide some context for our results, in terms of projected changes in precipitation seasonality, soil moisture etc., referring to Kay et al. (2022) which discusses soil wetting dates

General comments:

- There seems to be an issue with your referencing system. I found at least three references cited in the text to be missing in the reference list (Coles, 2001; Jiminéz Cisneros et al, 2014; and Paz et al, 2006). I did not check all of them, so there could be more. Furthermore, the reference list is not always sorted alphabetically (e.g. Robson et al and Rudd et al should be before Sayers et al) and some references do not start on a new line (e.g. Chen et al. ).
  - Understood and we will correct and check this at the typsetting stage.
- Data availability: What is EIDC?

- o EIDC is "Environmental Informatics Data Centre". We will edit this sentence.
- There are missing spaces in lines 201, 223, 225, and 227, and an "s" missing in asymptotic in line 218.
  - o We will correct this.
- Line 181: There seems to be a word missing after "widespread".
  - o The missing word is "events" which we will correct.

**References**

Berghuijs, W. R., Allen, S. T., Harrigan, S., & Kirchner, J. W. (2019). Growing spatial scales of synchronous river flooding in Europe. Geophysical Research Letters, 46(3), 1423-1428.

Blöschl, G., Hall, J., Parajka, J., Perdigão, R. A., Merz, B., Arheimer, B., et al. (2017). Changing climate shifts timing of European floods. Science, 357(6351), 588-590.

Brunner, M. I., Gilleland, E., Wood, A., Swain, D. L., & Clark, M. (2020). Spatial dependence of floods shaped by spatiotemporal variations in meteorological and land-surface processes. Geophysical Research Letters, 47(13), e2020GL088000.

Lavers, D. A., Allan, R. P., Villarini, G., Lloyd-Hughes, B., Brayshaw, D. J., & Wade, A. J. (2013). Future changes in atmospheric rivers and their implications for winter flooding in Britain. Environmental Research Letters, 8(3), 034010.

Bell V.A., Kay A.L., Jones R.G. (2009). Use of soil data in a grid-based hydrological model to estimate spatial variation 275 in changing flood risk across the UK. J. Hydrol. 377(3–4): 335–350. doi:10.1016/j.jhydrol.2009.08.031Chen Y, Paschalis A, Kendon E, Kim D, Onof C (2020). Changing spatial structure of summer heavy rainfall, using convection-permitting ensemble. Geophysical Research Letters, 48, e2020GL090903. doi: 10.1029/2020GL090903.

Bell, V.A.; Kay, A.L.; Cole, S.J.; Jones, R.G.; Moore, R.J.; Reynard, N.S., 2012, How might climate change affect river flows across the Thames Basin?: an area-wide analysis using the UKCP09 Regional Climate M. Journal of Hydrology, 442-443, 89-104

Kay, A. L. (2021). Simulation of river flow in Britain under climate change: baseline performance and future seasonal changes. Hydrol. Process. 35:14137. doi:10.1002/hyp.14137

Kay, A. L., Bell, V. A., Guillod, B. P., Jones, R. G., and Rudd, A. C. (2018). National-scale analysis of low flow frequency: historical trends and potential future changes. Clim. Change 147, 585–599. doi:10.1007/s10584-018-2145-y

Kay, A.L., Lane, R.A. and Bell, V.A. (2022). Grid-based simulation of soil moisture in the UK: future changes in extremes and wetting and drying dates. *Environmental Research Letters*, 17(7), 074029, doi:10.1088/1748-9326/ac7a4e

Lane, R. A., & Kay, A. L. (2021). Climate change impact on the magnitude and timing of hydrological extremes across Great Britain. Frontiers in Water, 71. doi:10.3389/frwa.2021.684982

Price, D., Hudson, K., Boyce, G., Schellekens, J., Moore, R.J., Clark, P., Harrison, T., Connolly, E., Pilling, C., 2012. Operational use of a grid-based model for flood forecasting. Proc. Inst. Civ. Eng. - Water Manag. 165, 65–77. doi:10.1680/wama.2012.165.2.65

Rudd, A. C., Kay, A. L., and Bell, V. A. (2019). National-scale analysis of future river flow and soil moisture droughts: potential changes in drought characteristics. Clim. Change 156, 323–340. doi: 10.1007/s10584-019-02528-0

---

## Author Response (AR1)

**Responses to reviewers**

Line numbers correspond to the original paper, not the corrected version. Responses are given in blue following each point.

**Reviewer 1**

**General Comments**

The overall concept of this paper is neatly done: a 12-member ensemble of baseline and future climate (12 km resolution) is input to a grid-based hydrological model (1 km resolution) to characterise the impact of climate change on flood events. The strength of the paper is in its focus on areal flood events, where the joint interaction between the factors that cause floods over a range of temporal and spatial scales is implicitly accommodated by the use of a gridded daily continuous simulation model. All inferences about changes to flood risk are made using 30-year sequences of daily floods, as derived from the 12-member ensemble of climate projections. Differentiating impacts by the areal extent and duration of floods of varying severity is novel, as is the exploration of possible changes in their spatial dependency.

There are, however, some aspects to this work which are potentially problematic, and these need to be addressed by further explanation and/or revision.

**Specific Comments**

The key issues that I am struggling with are as follows:

- It is difficult for a dynamically downscaled rainfall products to reproduce rainfall quantiles over the temporal and spatial (meso-) scales relevant to catchment flooding, and I was surprised to read (lines 62-63) that "due to the focus on … extremes rather than the whole regime in general" that no bias correction was applied. Bias-correcting projected extremes is as important, if not more important, than a central tendency measure. The rainfall-based simulation of floods is critically dependent on the correct representation of the frequency distribution of areal rainfalls, and I think it important to provide evidence that the frequency of areal rainfall extremes derived from the UKCP18 data compare reasonably well with observations. To this end, providing evidence that distributions fitted to n-day maxima extracted from UKCP18 (preferably for a range of areal extents relevant to the adopted spatial limits) are reasonably consistent with those based on observational data. I searched for any such evaluations in the Met Office documents (the citations provided for these need to be improved and corrected in the manuscript) but I could not find anything specifically relevant to the rainfall behaviour of most interest.

  - There are many issues and assumptions inherent in the bias-correction process, including the assumption that the same 'biases' seen in baseline climate model data are also present in data for future periods, concerns that correction can alter the spatio-temporal consistency of individual variables or break important physical relationships between variables, and the fact that typically-applied daily rainfall corrections can fail for multi-day rainfall totals (e.g. Ehret et al. 2012, Addor and Seibert 2014). The application of bias-correction can even introduce artefacts into the 'corrected' data (Maraun et al. 2017). Attempts to 'correct' rainfall extremes are especially difficult, as by their nature they have limited occurrence in observation-based datasets and are also strongly affected by natural climate variability (e.g. the well-known presence of prolonged flood-rich and flood-poor periods), whereas ensembles of climate model data will not necessarily present the same 'states' of natural variability through time. Thus application of bias-correction, rather than reducing uncertainty, represents a considerable source of uncertainty in itself (e.g. Lafon et al. 2013, Ehret et al. 2012). In this application we instead chose to use only the raw climate model data, to maintain the spatio-temporal properties of precipitation, temperature and potential evaporation imposed by the dynamic downscaling of the RCM. We then determine what constitutes an "extreme" level of flow by selecting a threshold based completely off the climate model runs, not observations and hence any bias in threshold selection is matched by bias in the events. The upshot of this is that the key features and results are not impacted by the bias.

- On the basis of the information provided it is difficult to be comfortable with the reported probabilities of exceedance (PoE). In concept the approach of adopting a merged CDF on the basis of empirical and fitted distributions is fine, my difficulty is with the inferred annual PoEs. I suspect that there is a problem with the way that the Poisson approximation is applied, and I suggest that the authors compare (or replace) their analysis with the more straightforward approach based on fitting the GPA distribution to the POT2 series, where the annual quantiles are obtained by the simple expedient of factoring the exceedance probabilities

by N/M, where N is the number of years in the record and M is the number of maxima extracted. The key reason for my discomfort with the PoEs reported is the severity of the identified events. For example, in Figure 2 it appears that 3 (possibly 4?) events with return periods of 1000 years have been observed in a single 30 year sequence. I appreciate the need to consider the influence of spatial dependency and the trading space for time issues here, but still, this number of extreme events is higher than expected (and higher than I suspect would be extrapolated by Tawn et al, 2019). A crude estimate of the likelihood of this could be obtained by estimating the notional number of largely independent catchments across the UK. If we adopt a spatial dependence limit of 120km (from line 220 in the paper) then the notional upper limit of the spatial extent of an event might be around 45000 km2, which yields around 5 or so independent catchments (or "trials") in each year. Given that the likelihood of a 1 in 1000 event occurring in a 30-year period is 0.029 (from the Binomial distribution), then there is about a 13% chance you would see a single 1000-year event in one of the five independent catchments somewhere across the UK in a 30-year period. However, we would actually need to have around 50 independent catchments in the UK to see three 1000-year events occurring in a 30-year period with any likelihood, and this corresponds to an asymptotic dependence limit of only around 40km, which is very low given the information presented in Figure 7. The number of exceedances shown in Figure 5 is larger again, but this may be due to how the ensemble members are combined (discussed in the next point).

- This is a really interesting point, and quite an insightful way of estimating the number of very extreme events within a given period. The merged CDF is required for the copula method to be applicable, however, the empirical component of the distribution is not actually used in the figures since the threshold for using the GPa exactly corresponds to our threshold for delineating the event extents. A preliminary investigation suggests that your alternative greatly reduces the return periods of the most extreme events, reducing most of the >1000 year return periods to under 1000 years. However, we have a second paper in publication building on this work, and the authors feel that a consistent presentation of return periods across the two papers would minimize confusion. As we feel this is an important point to make, we will include a paragraph at line 156 outlining the alternative approach.
- Line 156: "An alternative approach is to simply use $F_{GPA}(x)$ directly, and scale probabilities by N/M, where N is the number of years in the record, and M is the number of exceedances extracted. Due to the small probabilities involved, these don't line up for the largest values of flow. This approach leads to smaller estimates of return period, which might potentially align with discussion of the "frequency of 100-year events in the UK" (Tawn et al., 2019)."

- If my understanding is correct (lines 175-177), the 12-member ensemble from UKCP18 has been lumped together and used in the preparation of the results as summarised in Figures 3 to 7. I think this approach confounds the absolute interpretation of the reported frequencies and return periods, and I suggest that it would be more useful to treat each ensemble member as a source of aleatory uncertainty over a 30-year period. Thus, rather than reporting, say, that there are 17 events larger than 1000-year event in DJF (Fig 5) under baseline conditions, it would be more useful to report on the average (or median) frequency/quantile across the 12-member ensemble, where the highest and lowest ensemble member provides an indication of the upper and lower bounds of the sampling uncertainty in each 30-year period.

- This is a good point. Aside from Figure 3 which is already split by ensemble member, we can easily include uncertainty bounds on Figure 4, and include some measure of variance in Figures 5, 6 and 7 by including three supplemental figures showing how the spread of the distribution changes between ensemble members using convex hulls of the underlying points on each figure. Including upper and lower bounds in addition would result in a lot of extra figures, or much more complex ones, at a cost to readability. At line 177 we replace the sentence "In the rest of this section, the event sets …" is replaced by "In the rest of this section, the event sets from all ensemble members are combined and given equal weighting. In the supplementary material, ensemble members are treated as separate sources of equal weighting. Ensemble members are shown in different colours and have the convex hull of the points from each ensemble member highlighted to show in particular variation in the extremes."
- We also add three figures into a new supplementary material, corresponding to showing variance related to figures 5, 6 and 7.
- Line 200: "Supplementary material figure 1 shows that there is some variability between ensemble members, particularly in the extremes, but the overall pattern is preserved throughout, as expected from Figure 3."
- Line 206: "Supplementary material figure 2 shows that this pattern is matches between ensemble members, but there is some variability in the relative dynamics of duration and rarity in the extremes."

- o Line 227: "This is also mirrored in Supplementary material figure 3, which shows this split by ensemble member, where spatial variation in coherence is strongly preserved between ensemble members."
- Lastly, no discussion is provided on how the asymptotic independence metric varies with distance (lower panel, Figure 7). I think the metrics used by Coles to explore asymptotic behaviour would benefit from additional explanation here as they are not intuitively obvious; specifically, the way in which the independence metric is defined is easily misinterpreted and without explanation it appears odd that the degree of independence is decreasing with increasing distance, which is exactly the opposite of what one would expect (and as shown in the dependency metric in the upper two panels of Figure 7, which is consistent with intuition).
  - o This is a reasonable point to make. On the one hand, we do not wish to just repeat Coles, however we agree that intuition may be misleading. We edit the text at line 136 to the following: "…are calculated between pairs of points. For two points i and j,

$$\chi_{i,j} = \lim_{x \to \infty} P[Q_i > x | Q_j > x]$$

    If $C^*(u,v) = 1 - u - v + C(u,v)$ for a copula C between two points i and j, then

$$\bar{\chi} = \lim_{u \to 1} \frac{2 \log (1-u)}{\log C^*(u,u)}$$

    χ describes the level of asymptotic dependence, if χ = 0 then the variables are asymptotically independent, otherwise they are asymptotically dependent. In the asymptotically independent case, $\bar{\chi}$ describes the dependence for large but not asymptotic values of flow. In the asymptotically dependent case, $\bar{\chi}$=1."
  - o To comment on both panels of Figure 7, we change the sentence on line 220 to: "The figure suggests that asymptotic dependence decreases as distance increases. In the asymptotically independent case (Figure 7b), we see a similar pattern in dependence for large values of flow, with high dependence at short distances, even if they are independent in the limit."

**Reviewer 2**

The authors present a study on flood changes under climate change in the UK. The focus lies specifically on changes in modelled flood return periods based on an ensemble climate projection. The main point of analysis is the changes in widespread flooding. The authors find that there is more widespread flooding in winter and less in summer in the future projected climate. Further analysis included changes in return period, area covered and duration of events between current and future climate.

Overall, I like that the article focuses specifically on simultaneously occurring flood events under climate change. The analysis and results presented here show thorough, good work. I would have wished for a bit more focus on how the uncertainty of the climate ensemble translates into the results and more discussion of the results regarding potential drivers of change. While I have lots of comments and open questions, all of them are minor and should be quick to address.

**Introduction**

- Since a large part of your results section talks about spatial dependence, can you motivate this analysis in the introduction? Especially since several people have already written about flood coherence/synchrony (Brunner et al. 2020) including results for the UK (Berghuijs et al. 2019).
  - o On line 39 we add the following: "… more frequent and severe. Spatial coherence of flooding events – whether flood timings at different locations have become more correlated – is of key interest to national-scale actions to mitigate the associated loss. The dependence structure of river flow has been analysed on a Europe-wide scape (Berghuijs et al., 2019) and focusing on the United States (Brunner et al., 2020), focusing on synchrony of events within a given range."

**Methodology:**

- Can you elaborate on why you chose the Grid-to-Grid model for this analysis and how well it performs in streamflow/flood prediction under the current climate? This would allow some estimate how reliable future projections might be.

- o We will add some further information to line 74 about previous studies using Grid-to-Grid: "Grid-to-Grid has been widely tested and applied to explore climate change impacts on river flows across GB, for both floods (Bell et al., 2009, 2012; Kay et al., 2018) and droughts (Rudd et al., 2019; Kay, 2021; Lane and Kay, 2021). It has also been used by the English Environment Agency for flood forecasting (Price et al., 2012). "
- I like that you thoroughly elaborate on your choice of thresholds regarding POT and inundation extend. Can you supplement this with a sentence along the lines of "Widespread events are defined as…".
  - o Agreed. We add the following at line 95: "We define widespread events as timepoints for which a large number of locations experience very high flow (i.e. above the POT threshold) simultaneously."
- Can you elaborate more on the method chosen for asymptotic dependence and, more importantly, elaborate on what that means? I have not encountered this method before, nor did I understand by the end of the paper, what it actually tells me. If you are interested in using an established method, I can refer you again to the papers by Brunner et al, (2020) or Berghuijs et al. (2019). Their results should also be discussed in line 254 since it relates to your proposed further work.
- o The measures of asymptotic dependence are well established but often misunderstood. To alleviate this, We edit the text at line 136 to the following: "…are calculated between pairs of points. For two points i and j,

$$\chi_{i,j} = \lim_{x \to \infty} P[Q_i > x | Q_j > x]$$

If $C^*(u,v) = 1 - u - v + C(u,v)$ for a copula C between two points i and j, then

$$\bar{\chi} = \lim_{u \to 1} \frac{2 \log (1-u)}{\log C^*(u,u)}$$

χ describes the level of asymptotic dependence, if χ = 0 then the variables are asymptotically independent, otherwise they are asymptotically dependent. In the asymptotically independent case, $\bar{\chi}$ describes the dependence for large but not asymptotic values of flow. In the asymptotically dependent case, $\bar{\chi}$=1."
  - o To comment on both panels of Figure 7, we change the sentence on line 220 to: "The figure suggests that asymptotic dependence decreases as distance increases. In the asymptotically independent case (Figure 7b), we see a similar pattern in dependence for large values of flow, with high dependence at short distances, even if they are independent in the limit."

  - o On line 254, we add the following: "…event length determination. Brunner et al. (2020) make use of a spatial dependence function (F-madogram) and hierarchical clustering to determine events for which points are mutually dependent to a sufficient degree. This would be an interesting direction to go in to improve event identification. To highlight spatial dependence in a simpler way than χ, Berghuijs et al. (2019) use a metric of flood "synchrony", measuring how often extreme floods occur at the same time within a given radius of a target point. The gridded data set we have available here could be evaluated using this metric, or one like it."

**Results:**

- There seems to be a mix between results and discussion in the results section (e.g. lines 184-190 are discussion, not results). You could either call the results section "Results and Discussion" or move any discussion from the results section to "Discussion and Conclusion". Generally, the discussion could be more elaborate (see below).
  - o We call it Results and Discussion.
- You quite often talk about an "increase in the range", "little change", "less asymptotic dependence", "extend slightly". Can you support these statements with numbers?
  - o We add percentage changes where appropriate throughout the results and the discussion, and add change in χ and χbar as well.
  - o Line 197: "In the future time-slice, event duration seems to be slightly shorter on average, and this is more pronounced in spring (MAM) and summer (JJA) , reducing from 3.54 to 2.99 days in spring, 2.20 to 2.04 in summer."
  - o Line 202: "The changes between the two time-slices are subtle, but there is an overall trend towards an increase in the range of peak return period: the 95th percentile of return periods increases in all

seasons, from an increase of 10 years in spring to 205 in summer, with the 5$^{th}$ percentile being ~1.2 in all seasons and timeslices."

  o Line 220: "There seems to be little change in dependence between the two time-slices, although the asymptotic dependence appears to extend slightly further in the present time-slice (a maximum distance for which $\bar{\chi} = 0$ of 300km in the present versus 260km in the future). If the events are subdivided by season, subtle differences can be observed (Fig 8). Overall, spring and summer shows less asymptotic dependence (lower values of $\chi$ and $\bar{\chi}$) than autumn and winter. In spring and summer, mean $\chi$ is 0.641 and mean $\bar{\chi}$ is 0.222, compared to mean $\chi$ is 0.673 and mean $\bar{\chi}$ is 0.363 for autumn and winter. Also, the 50% contour for $\bar{\chi}$ is longer in spring (max distance of 495km in present, 545km in future) than summer (max distance of 431km in present, 462km in future) in both time-slices, suggesting that the variance in $\bar{\chi}$ exhibits seasonal variation."

- Line 205: "On the right of some panels (future winter and autumn) is a set of events with a peak return period of at least 1000 years." From what I see, all panels have events up and over a return period of 1000 years.

  o We change this sentence to "We see that in all seasons are a small number of events with return periods exceeding 1000 years, particularly in winter and autumn."

- Even though you use a climate ensemble as input data for the hydrological model, the presented results mostly do not give an overview of the uncertainty the different climate projections introduce. Can you please give an indication of how the ensemble spread demonstrates uncertainty in the results? Especially since you state in the abstract: "Results were consistent across ensemble members, with none showing significant difference in distribution." Since the two main conclusions are about the seasonal shift and spatial dependence, the results in Figure 3 are not enough to support this statement across all findings.

- Aside from Figure 3 which is already split by ensemble member, we can easily include uncertainty bounds on Figure 4. On line 186 we add: "Between ensemble members, variability is higher in Summer, even more so in the future time-slice, and one member actually saw an increase in summer events in the future time-slice (Figure 4, right)." A second panel is added to figure 4 to describe variability between ensemble members.

[Figure]

**Figure 1 Number of widespread events, summed across ensemble members, split by season and time-slice. Left: total events, Right: mean number of events per ensemble member with error bars showing minimum and maximum across ensemble members.**

- Including upper and lower bounds in addition would result in a lot of extra figures, or much more complex ones, at a cost to readability. At line 177 we replace the sentence "In the rest of this section, the event sets …" is replaced by "In the supplementary material, ensemble members are treated as separate sources of equal weighting. Ensemble members are shown in different colours and have the convex hull of the points from each ensemble member highlighted to show in particular variation in the extremes.

- Figure 2: Can you include in the caption what the percent inundation refers to? Is this percent grid cells or percent land area?

  o We add the following on line 160: "The percentages show refer to the percentage of the number of river grid cells, not a fraction of UK land area."

- Figure 3: I would prefer if you would present a summary figure for the different model ensembles. After all, since the ensemble runs represent uncertainty, only presenting, comparing and analysing individual ensemble members does not make sense.
  - As mentioned above, we added Figure 4, and three supplementary figures. However, Figure 3 is important to highlight that we have not erroneously included ensemble members which differ significantly from the others. The specific statistics are covered more broadly in the subsequent figures, so we feel adding to this information-heavy figure would reduce readability.
- Figure 4: Since you are using ensemble results, can you include uncertainty bars into the event count? Secondly, the caption says that you take the sum of all ensemble results. I would think that the mean or median (and potentially even the range) is the more appropriate measure. This is the case for Figures 5 and 6 as well.
  - See above for our response on this

- Figure 5+6: Is there a specific reason why you have return period once on the x-axis and once on the y-axis? If not, I would recommend choosing one or the other, not both.
  - This is a formatting choice. In Figure 5, duration is much narrower in range than return period, so the heat maps have a "vertically" stretched profile. Conversely, the key features in figure 6 vary more by return period, and so again, due to the number of panels, the order was chosen to keep the figures clear.

**Discussion:**

- Although the analysis itself does not focus on drivers of change, there have been several published articles on how hydrology and specifically floods are changing in the UK. I think the discussion would benefit from discussing the results of this study in the context of previous findings. For example, there is a projected increase in winter atmospheric rivers in the UK which are likely to bring widespread flooding (Lavers et al, 2013). Furthermore, floods in the UK are strongly associated with soil moisture timing (Blöschl et al, 2017). Do changes in the soil moisture influence in the increase/decrease of widespread flooding in the UK?
  - We will add a couple of sentences to the discussion (at line 244) to provide some context for our results, in terms of projected changes in precipitation seasonality, soil moisture etc., referring to Kay et al. (2022) which discusses soil wetting dates
  - Line 239: "This matches with some work done by Lavers and Villarini (2013) which shows the possible increase in atmospheric rivers, especially in Western Europe, which drive extreme precipitation events. The typical spatial extent of events was found to be fairly consistent between time-slices, but summer (June-August) events appeared smaller in the future across all return periods. Event duration decreased on average in all seasons between the two time-slices. This pattern was the same across all RCM ensemble members. Kay et al (2022) show that projections of soil moisture changes point towards wetter winters and drier summers. In conjunction with Blöschl et al (2017) suggesting that UK floods are closely linked to soil moisture timing, this gives confidence in the results in the present work"

**General comments:**

- There seems to be an issue with your referencing system. I found at least three references cited in the text to be missing in the reference list (Coles, 2001; Jiminéz Cisneros et al, 2014; and Paz et al, 2006). I did not check all of them, so there could be more. Furthermore, the reference list is not always sorted alphabetically (e.g. Robson et al and Rudd et al should be before Sayers et al) and some references do not start on a new line (e.g. Chen et al. ).
  - These, and those mentioned throughout these responses have been edited in the references.
- Data availability: What is EIDC?
  - EIDC is "Environmental Informatics Data Centre". We have edited this sentence.
- There are missing spaces in lines 201, 223, 225, and 227, and an "s" missing in asymptotic in line 218.
  - We have corrected this.
- Line 181: There seems to be a word missing after "widespread".
  - The missing word is "events" which we have corrected.

**Editors remarks**

This is an interesting methodology with a focus on spatial floods. The reviewers have made useful suggestions, and from what I can read in the responses they are addressed well.

Additionally, I suggest that the authors emphasise the methodological aspects of the paper and downplay the findings of the simulations. The findings are surely of interest to flood managers in the UK, but HESS is an international journal, and for an international audience the methods are of much more relevance.

- This point around methodological relevance is indeed important to an international audience. We feel that the addition of the supplementary material, and the embellishment of several methodological parts (such as the definition of asymptotical dependence, and the issue of using a Poisson approximation for daily-to-yearly exceedance probability conversion) increases the emphasis on the methodological. The authors are somewhat reticent to change too much more as there is already a second paper (Adam Griffin, Alison Kay, Elizabeth Stewart, Paul Sayers; Spatially coherent statistical simulation of widespread flooding events under climate change. Hydrology Research 1 November 2022; 53 (11): 1428–1440. doi: https://doi.org/10.2166/nh.2022.069) which focuses on the methodological. We feel the abstract and conclusions do a lot to emphasise the method over the UK-centric implications, and demonstrate its applicability to a global stage.

Please submit a document detailing how you have addressed all comments along with the revised paper.

**References**

Berghuijs, W. R., Allen, S. T., Harrigan, S., & Kirchner, J. W. (2019). Growing spatial scales of synchronous river flooding in Europe. Geophysical Research Letters, 46(3), 1423-1428.

Blöschl, G., Hall, J., Parajka, J., Perdigão, R. A., Merz, B., Arheimer, B., et al. (2017). Changing climate shifts timing of European floods. Science, 357(6351), 588-590.

Brunner, M. I., Gilleland, E., Wood, A., Swain, D. L., & Clark, M. (2020). Spatial dependence of floods shaped by spatiotemporal variations in meteorological and land-surface processes. Geophysical Research Letters, 47(13), e2020GL088000.

Lavers, D. A., Allan, R. P., Villarini, G., Lloyd-Hughes, B., Brayshaw, D. J., & Wade, A. J. (2013). Future changes in atmospheric rivers and their implications for winter flooding in Britain. Environmental Research Letters, 8(3), 034010.

Bell V.A., Kay A.L., Jones R.G. (2009). Use of soil data in a grid-based hydrological model to estimate spatial variation 275 in changing flood risk across the UK. J. Hydrol. 377(3–4): 335–350. doi:10.1016/j.jhydrol.2009.08.031Chen Y, Paschalis A, Kendon E, Kim D, Onof C (2020). Changing spatial structure of summer heavy rainfall, using convection-permitting ensemble. Geophysical Research Letters, 48, e2020GL090903. doi: 10.1029/2020GL090903.

Bell, V.A.; Kay, A.L.; Cole, S.J.; Jones, R.G.; Moore, R.J.; Reynard, N.S., 2012, How might climate change affect river flows across the Thames Basin?: an area-wide analysis using the UKCP09 Regional Climate M. Journal of Hydrology, 442-443, 89-104

Kay, A. L. (2021). Simulation of river flow in Britain under climate change: baseline performance and future seasonal changes. Hydrol. Process. 35:14137. doi:10.1002/hyp.14137

Kay, A. L., Bell, V. A., Guillod, B. P., Jones, R. G., and Rudd, A. C. (2018). National-scale analysis of low flow frequency: historical trends and potential future changes. Clim. Change 147, 585–599. doi:10.1007/s10584-018-2145-y

Kay, A.L., Lane, R.A. and Bell, V.A. (2022). Grid-based simulation of soil moisture in the UK: future changes in extremes and wetting and drying dates. *Environmental Research Letters*, 17(7), 074029, doi:10.1088/1748-9326/ac7a4e

Lane, R. A., & Kay, A. L. (2021). Climate change impact on the magnitude and timing of hydrological extremes across Great Britain. Frontiers in Water, 71. doi:10.3389/frwa.2021.684982

Price, D., Hudson, K., Boyce, G., Schellekens, J., Moore, R.J., Clark, P., Harrison, T., Connolly, E., Pilling, C., 2012. Operational use of a grid-based model for flood forecasting. Proc. Inst. Civ. Eng. - Water Manag. 165, 65–77. doi:10.1680/wama.2012.165.2.65

Rudd, A. C., Kay, A. L., and Bell, V. A. (2019). National-scale analysis of future river flow and soil moisture droughts: potential changes in drought characteristics. Clim. Change 156, 323–340. doi: 10.1007/s10584-019-02528-0

**Additional References**

Addor N and Seibert J (2014). Bias correction for hydrological impact studies – beyond the daily perspective. Hydrol. Process. 28, 4823–4828.

Ehret U, Zehe E et al. (2012). HESS Opinions "Should we apply bias correction to global and regional climate model data?" Hydrol Earth Syst Sci, 16, 3391–3404.

Lafon T, Dadson S et al. (2013). Bias correction of daily precipitation simulated by a regional climate model: a comparison of methods. Int. J. Climatol. 33: 1367–1381

Maraun D, Shephard TG et al. (2017). Towards process-informed bias correction of climate change simulations. Nat Clim Change, 7, 764–773.

---

## Referee Report (RR1)

General Comments

The motivation of the paper to examine the spatial and temporal coherence of flood risks is well justified and the authors' approach to examining this topic is efficient and effective. However, the justification for using raw climate data for examining floods resulting from extreme climates is not yet satisfactorily justified, but I believe an additional validation step would allow the authors to retain the approach they have already adopted. I find the manuscript to be overall well written and figures are mostly well presented (some improvements are needed for Figs 3, 4, 7, and 8 and Figure 5 needs to be revised). A number of corrections and clarifications are necessary to ensure that: consistent terminology is used; the language considers the global readership of this journal; descriptions are precise and objective; figures are easily interpreted; and referencing is accurate.

**Major comment**

I concur with the authors that the use of bias correcting in an attempt to capture rainfall extremes relevant to floods is difficult and introduces considerable uncertainty and the justification here is to avoid the introduced uncertainty and instead examine the differences in floods resulting from modelled historical and future climates. However, the justifications currently provided are inadequate and at times illogical.

The authors state that "due to the focus on [sic] the present work on extremes rather than the whole regime in general, bias correction is not applied here."
I'm afraid this justification is unsatisfactory. I'm certain the authors are knowledgeable of the fact that regional climate models are aimed at resolving large scale spatial and temporal variability, namely, what is referred to here as "the whole regime in general". A study that was aimed at large scale changes would therefore be well justified in using the raw RCM outputs. In contrast, extreme precipitation events are typically driven by synoptic scale events, which would justify the use of bias-corrected projections. Furthermore, flooding is sensitive to the spatial and temporal distribution of storms at even finer scales and have the added complexity of local factors that influence the flood response other than climate. The focus on extreme events is therefore a justification for introducing uncertainty to address the increased complexity of flood responses. Lines 65-66 therefore needs to be removed.

In their response letter, the authors make the case that they have kept the biases constant by comparing modelled historical and modelled projected floods and by capitalising on the dynamic nature of the GCMs. (however, it then follows that the same justification would not rule out the comparison of bias-corrected data – here too, the biases would be consistent between modelled historical and projected climates. Furthermore, bias corrected rainfalls are not constrained to the observation based datasets, as it appears to be suggested in the authors' response– both statistical and dynamical downscaling approaches allow for estimates outside the range of the observed records.)

This justification is sufficient and acceptable conditional upon the authors adopting the following suggestions, which should be easy to implement:

1. Repeat the analysis using observational data over the same period as the historical modelled data to quantify the degree to which the spatial and temporal coherence

of flooding is approximated and represented in the baseline case. This would then provide evidence of the fidelity of the baseline to which the *relative* changes in the number of widespread events could be estimated using the approach that has already been presented. Currently, the presentation of results dependent only on modelled climate data makes it easy for a reader to suppose that the results reflect similarities in catchment characteristics combined with the lack of spatial specificity in the modelled climate date. Having an observation-based analysis would provide context. Presenting these observed results in the SI would be adequate

2. Present the findings as relative changes in the number and spatial scale of the widespread events (more details of this in reference to figure 3 below)

**Specific comments (line numbers reference manuscript 2)**
Acronyms are often used before they are defined – the ones I noticed were: NRFA, RCM (this shouldn't be assumed knowledge since GCM is previously defined), PoE. There may be others.

It would be helpful to be clarify the terminology used to describe seasons - this is primarily for your readers in the southern hemisphere, please don't neglect us. Please consider including "boreal" when seasons are referenced. Alternatively, include the months in brackets after each season. It may seem superfluous for a northern hemisphere native, but it makes interpreting the results so much easier for those in the south.

The time slices are introduced as a "baseline" and "future", but the former is later referred to as "present". Please keep the terminology consistent, particularly since "present" is inaccurate when referring to a time period covering 1980-2010.

L 90: The term "percentiles" in reference to floods is used to describe distributions. The term that should be used here is *frequencies*, not *percentiles*.

L95: similarly, the notation $Q_x$ is widely used in hydrology to denote the x percentile of flow rather than a flood frequency. Please keep the terminology you have chosen consistent by referring to the 1 in 5 and 1 in 10 year probability event as POT 0.2 and POT 0.1 respectively. Alternatively, use the term Average Recurrence Interval of 5/10 years (ARI5 and ARI10).

L99: This is ambiguous as absolute values of flow are in fact being used as thresholds, and these thresholds are dependent on the distribution of the data: the definition of the thresholds are just location dependent. I suggest phrasing this as:
Note that the thresholds are not based on universally applied fixed values of flow magnitude but are instead dependent on thresholds defined by empirical flood frequencies.

L150: I had to read this several times and I'm still unclear. The text implies that $\bar{\chi}$ applies to large flows and conversely that $\chi$ applies to flows of all magnitudes (which I do not believe is the intended message). I suggest the following (but I'm unsure whether I've interpreted the text correctly. Please revise as necessary)
$\chi$ describes the level of asymptotic dependence; if $\chi \neq 0$ then the variables are asymptotically dependent. A value of $\chi = 0$ represents asymptotic independence. Asymptotically independence is also represented by a value of $\bar{\chi} \neq 1$. A value of $\bar{\chi} = 1$ represents flows that are dependent but nor asymptotic.

L185: please justify why these four events are selected. Why "four of" and why not "the four largest"? Are they selected to demonstrate that most events are spatially contiguous? (As an aside to the major comment above, is the spatial coherence an attribute derived from the modelled climate data, or is the pattern present in observed data?)

Figure 2: If I'm understanding this correctly, this figure shows the degree of spatial inundation with the colour scale showing the equivalent severity of the flood for events identified using the POT2 threshold, equivalent to a return period of 0.5. If this is the case, why are more frequent events with return periods of 0.2 to 0.5 shown? Perhaps they are not, but the light yellow scale is difficult to distinguish between greater or less than 0.5. I suspect it is simply a case of the legend needing to be updated to show grey between 0.2 and 0.5. Also, the text size of the labels on this and the next figure are rather disproportionately large. Please reduce the text size.

Figure 3: Comparing the changes between the baseline and future is not easy with the way this figure is presented – there are a lot of vertical bars in different ensemble members to compare and lining up the number of events for different return periods is challenging, while the colour selection implies that A and B are showing a different variable to C and D. I strongly suggest combining the information from A and C, and B and D by showing these results as the difference between time slices. If there is a very good reason to not do this, please at the very least change the colours to paired colours: e.g. light blue for A, blue for C, light green for B and green for D, so that light colours correspond to the baseline, darker colours to the future, blues for area, and greens for return period.

L 203-2007: The choice of using the word "may" makes the sentence sound speculative. These sentences could be revised to reflect the certainty of the results. Suggest the following or similar:
However, the increase in widespread events is confined to the boreal autumn (SON) and winter (DJF) with a decrease in events between March and August. The decrease in future boreal spring and summer events could be due to overall projections of drier summers (Murphy et al., 2019), or could result from a spatial contraction of summer floods in the future, which have historically resulted from short-duration, high intensity storms.

L 212: It's unclear what the term "methods" is in reference to. Is it the method of climate modelling or is this meant to mean differences in flood responses to different storm types?

Figure 4: The left figure is superfluous. Please remove this.

L227-230: Describing the changes in reference to changes in return periods is unconventional and is probably due to the way the figure axes are configured (see comment on Figure 5). Please change this description to one that describes the shift in event duration with respect to frequency.

Figure 5: is there actually a heavier tail in the future in SON? (L223)
Also, the return period *must* be plotted on the x axis, as the duration of the event is dependent on the return period not the other way around. This isn't just a formatting choice – the reader is forced to attempt to transpose figure 5 as the convention in referring

to tails is the distribution along the horizontal (with a few exceptions). In addition, there isn't a good reason to present Figures 5 and 6 in a way that is disjointed. The reason given in response to the previous reviewer of duration having a smaller range than return period (and likewise in Figure 6 the return period having a smaller range than area) is disputable: one could easily represent the return period in log10 years and have an effective scale of 0-4 (just making a point – I'm not suggesting that return periods be presented in this way).

Figure 7: labels a, b, c, d are missing from Figure 7 (b is referenced in the text). The caption needs to describe what is in each of the four figures (i.e. different time slices and different measures). This figure is also inconsistent with previous figures that have shown the baseline time slice on the top row and the future on the bottom (this can be fixed and the legends could simply be placed horizontally under the respective columns). Using a different color scheme for each metric would aid in interpreting the different scale of results.

Figure 8: as per Figure 7 regarding layout and caption. Only one legend is needed.

**Minor comments**
Grammar and style
L66: "of the present work"
L175: "shown" instead of "show"
L231-233: missing "there"; "matched" not "matches"; "dynamics" is not an appropriate adjective here. I think what is meant is: but there is variability amongst the ensemble members in the relationship between event duration and frequency.

L237: "is not surprising": please replace this with objective language e.g. statistically plausible

Referencing: Following on from a previous reviewer's comment, please check the accuracy of all references. One example is that of Tawn et al. 2018: it is referenced as 2019 in the text and the list of authors in the reference list is incorrect. The dates for Towe are also reference incorrectly in text. There may be others, so it may be prudent to check the referencing system.

---

## Referee Report (RR2)

**General**

As stated previously I do think the strength of the paper is its use of a climate model ensemble to simulate areal flood events, where the joint interaction between the factors that cause floods over a range of temporal and spatial scales is implicitly accommodated by the use of a gridded daily continuous simulation model. I have gone through the various comments and responses, and overall I appreciate and agree with the changes made by the authors.

I appreciate that the authors have undertaken a computationally demanding set of simulations – the data sets and modelling framework are impressive – but I regret that I am left with concerns about two major points which I raised previously, namely 1) the need to compare baseline modelling with observations, and 2) the defensibility of the probability of exceedance estimates. The authors' response to the former point focussed on the problems with bias correction without addressing the more fundamental concern around the need to demonstrate how well the modelled baseline frequency curves of areal rainfalls or floods conform to observations. The response to the latter point (the method of calculating probabilities of exceedance) does not provide additional confidence in the validity of the results and some reconciliation of the different estimates is needed.

I provide more commentary on these two points below.

**Baseline Evaluation**

I agree with the authors' caveats about the dangers of bias correction, but my main point in this regard was the need to provide evidence that the frequency distribution of areal event extremes derived from the UKCP18 data compare reasonably well with observations. My earlier comments expand on this point a little, but as far as I can tell the authors' justification for not examining such evidence is because they adopted flow thresholds to yield a specified number of flood exceedances over a given period (and over a given area). This approach does not demonstrate how well the results generated over the baseline period relate to real-world conditions, it is merely a device to extract the number of events relevant to the exceedances of interest. What is missing here is a comparison of the selected thresholds with what has been observed over some suitable baseline period.

In other words, the selection of thresholds to yield a defined number of flood occurrences provides no information on how well the magnitude-frequency relationship of the flood regime is preserved (ie how well the cumulative density function governing the extremes matches reality), and this is a particular problem as the modelled floods are fitted to a Pareto distribution and the results are reported on in terms of absolute shifts in return periods (ie in terms of a shift in the magnitude-frequency relationship).

The results derived by fitting a Pareto distribution to the modelled events are very likely to be impacted by various forms of bias, and varying the thresholds to achieve the required number of exceedances does not avoid or obviate the need to undertake bias correction, it just ignores the problem. The degree of difference between modelled and observed flood

frequency curves for some representative locations would add considerable weight to the conclusions; without demonstrating this, the conclusions would need to state clearly that the adopted methodology yields results that have not been evaluated against observations over the baseline period, and thus are rather more speculative than currently framed.

**Probabilities of Exceedance**

I also think we should be troubled by the difference in results obtained using the two procedures to estimate the probabilities of exceedance (ie the one based on deriving annualised exceedances from a fitted GPA distribution with an assumed spatial dependence, and the method as described in the paper).

The authors have stated that they wish to retain the method as outlined in the paper as it is being used in another manuscript and they wish to "minimise confusion". While I appreciate the convenience of this decision, if the method cannot be shown to be correct then I see little point in being consistent.

The problem is that we have two methods (applied with different levels of rigour) that have two markedly different risk implications: I suggest that the subsequent check made with the assumed degree of spatial dependence yields results that are consistent with expectations, whereas the results presented in the paper are not. I think this a problematic outcome.

Accordingly, I think additional investigation is required to reconcile these different probability of exceedance estimates. The differences in results are too great and too surprising to ignore, and some diagnostic checks should be devised to check whether the inconsistencies are due to computational errors or unsupportable assumptions in one or other of the approaches.

Rory Nathan
University of Melbourne

---

## Referee Report (RR3)

The paper investigates the change to wide-spread flooding in terms of number of occurrences duration and extend between current and future climate using an ensemble regional climate model. This is a very relevant study showing novel results. However, revisions are required as per the comments below.

Main comments

The novelty and aim of the paper is not clear after reading the introduction. In the introduction the relevance of the work is clearly discussed, however no clear research gaps are mentioned. Also, at the end of the introduction (L46-50) a short summery of the methods rather than the aim of the paper is provided.

Some clarifications are needed in the data and methods sections (see specific points below). Especially, the statement that bias correction of precipitation from climate models is not required when focusing on extremes needs explanation. Furthermore, I think if the sensitivity of the assumptions (event magnitude threshold, extent threshold, and maximum event duration) in section 3.2 should be discussed separately from the methods as the current structure makes it hard to follow the methodology. And the manuscript would really benefit from a discussion on how these assumptions affect the final results (rather than the number of events for a single ensemble).

The conclusion section also contains discussion of the results as well as some limitations. For clarity, I recommend splitting the conclusions and discussion by making a separate section on limitations and moving the discussion of the results in relation to other papers to the "results and discussion" section.

Specific comments

L12: "…, allowing events to last up to 14 days" seems to suggest that the 14 days are a consequence of the event definition given before, but I don't see how this is the case. Can you clarify?

L36: I suggest to leave out "driving" in "driven by large ensembles of driving data from climate models" as it seems double?

L37: Can you provide more information about what "predominantly stochastic event-based models are"?

L42: Replace "focusing on the United states" by e.g. "for the United States", as using "focusing" twice in this sentence is confusing

L54: This is the first time UKCP18 is explained while it has already been used in the abstract and introduction. Please explain at the first use.

L66: The authors seem to state that bias correction of precipitation from climate models is not required when focusing on extremes? I would disagree with this and would like to see more better argued why no bias correction is required.

Section 2.1: Why was RCP8.5 selected? It would be good to mention this choice also in the abstract.

Section 3.1 It would be good to add which version of the model is used. Whether the model has been calibrated for this study or a previously calibrated version has been used.

L105: "this was considered equivalent ….". This subsentence is confusing to me. Please consider leaving it out or clarify its meaning.

Table 1: "PoE" is not explained. Furthermore, it would be clearer if consistent naming of thresholds were used in the "exceedances" column and in the text (e.g. POT2 or 2/yr). Also note the typo in "exceedances".

L119: I don't understand why based on minimum-threshold, flood extents that are smaller than this threshold are "retained". I would expect these are excluded. Could you please clarify?

L154: Why were 60 peaks selected and how does this relate to earlier event magnitude threshold? Also, from which ensemble or for all ensembles? Please clarify.

L159: How is the "daily exceedance probability" calculated from 60 peaks? I'm used to converting these probabilities to annual exceedance probabilities directly based on the average number of peak events per year, which you seem to refer to as an "alternative approach" (L171).

L174: Could you provide more context to the sentence "which might potentially align with discussion of the frequency of 100-year events in the UK"? What is this discussion about?

Section 4: In Figure 3, 5, and 6 event return periods are shown (if I understand correctly). It is however unclear to me how these are calculated? As show in Figure 2 the return period varies spatially, and it is not clear how a single return period is calculated.

L234: I assume AMAX is "annual maxima"? This has not been explained before.

L237: What do you mean by "change in flow"? Is that change in peak event magnitude?

L247: Can you explain what the value of 120 km is based on? In the figure it seems there are still points whit significant asymptotically dependence up to 250 km.

L271: I would be very useful to understand how sensitive the change in number of events is to the selection of thresholds in section 3.2. And how significant it is given the differences between ensemble members.

L299: Can you clarify what you mean by surface water flooding as opposed to fluvial flooding?

---

## Referee Report (RR4)

**Overall comments and recommendations**

The authors have given this revision of the manuscript considerable thought and effort, and the additional material helps to highlight both the challenges and value of the work undertaken. I raised two major concerns previously, namely 1) the need to assess the efficacy of the model simulations against suitable reference data, and 2) the manner in which the probabilities of exceedance were estimated. I am comfortable with the changes made to address the latter concern, so my comments below are focused on the efficacy of the adopted modelling chain. I sat on my draft comments for a week before finalising as I would like to see the paper published and I appreciate that the authors will be getting frustrated with this review process; however, I am struggling to agree with the author's interpretation of the new evidence presented and to date only the RCM projections and not the G2G model itself has been assessed.

The key contribution of the paper rests on characterising the possible changes to areal flood behaviour under climate change, where the adopted method implicitly accounts for the joint interactions between soil moisture, rainfall, and the non-linear influence of increasing catchment scale. The work leverages a large body of prior work undertaken to provide the adopted regional climate model (RCM) ensembles and G2G (hydrologic) modelling; while the use of this inherited work is of great benefit, it also brings with some specific challenges relevant to the stated research objective.

My interpretation of the new information provided in the last revision is that the RCM ensembles do a poor job of simulating the temporal and spatial correlations of the key processes of interest, and this undermines the core conclusions of the paper as currently framed. It also raises questions about the defensibility of the G2G simulations. While it is perhaps unreasonable to expect that the authors somehow "fix" deficiencies in the prior modelling effort that they have inherited, I do think it important that any shortcomings in the modelling chain are explored and used to qualify the conclusions made and to highlight areas for future research.

It appears that my assessment of the new information differs from the stated views of the authors, and this may be because I am misinterpreting information presented in the manuscript. So, either my concerns point to the need for the authors to better clarify and justify their reasoning, or else to perhaps revise their views based on the comments below.

I think the contribution of the paper would be greatly enhanced if the authors:

1) replace the current Figure 1b with an equivalent figure based on G2G results forced by "observation-based simulations" (rather than by the current 12-member RCM ensemble), and replace Figure 1a with a simple scatter plot of observed vs simulated Q50; the former plot provides insights about spatial biases related to regional hydroclimatic differences, and the latter plot about the nature of any overall model biases.

2) adopt a more critical approach to reviewing the implications of the results from the proposed G2G assessment as described in the previous point, and also from the RCM ensemble assessment already undertaken and currently summarised in Figures 4, 6 and 7.

The effort involved in undertaking 1) should be modest as the necessary hydrologic simulations have already been undertaken, but if this is not possible for some reason, then the manner in which the discussion and conclusions are written could be revised to better highlight, or refute, the perceived limitations in the hydroclimatic projections on which the main conclusions are based.

Further rationale for these recommendations is provided below.

**RCM projections**

Comparing the 1981-2010 flows derived from the RCM ensemble with those from the "observation-driven" inputs provides a very effective evaluation of the modelled climate over the 1981-2010 period. It is more valuable than comparing any individual climate variable alone (e.g. rainfall) as it implicitly allows for the joint distribution of climatic factors that influence floods, so this is a very useful addition to the paper.

However, at present the authors conclude that the RCM simulations of event areas are "fairly consistent" with those derived using SIMOBS, and that there is "slight bias" in the distributions of event severities. I find it difficult to accept these conclusions as stated because:

> 1) the differences in results between the SIMOBS and the mean RCM ensemble simulations as shown in Fig 4 are appreciably larger than the modelled differences between the climatic baseline and future conditions, and

> 2) the SIMOBS results appear to lie outside the maximum and minimum range of individual ensemble RCM results over the period 1981-2010 as deduced from a comparison between Fig 4 of the MS and Figure 1 of Supplementary material.

The differences in results would be better illustrated by including 'error bars' in Fig 4 that show the min/max spread of ensemble results for the 1980-2010 period, or else illustrating these differences in some other more easily comprehended fashion. It would also appear that the RCM simulations greatly overestimate the duration of the events (Fig 6), where it is observed that the differences between the RCM ensembles and the SIMOBS results are considerably greater than the projected differences due to global warming. The simulation of flood extents (Fig 7) appear more reasonable (but there is still an order of magnitude difference between simulated and observed maxima), and I agree with the interpretations of the authors regarding the adequacy of the seasonality of the flood regime as shown in Fig 5.

In short, the results shown in Figures 4, 6 and 7 appear to suggest that the RCM climate ensembles overestimate the spatial dependence of rainfall events and over-estimate the serial dependence in the correlation structure of rainfall extremes, and further that the distribution of modelled areal maxima is perhaps not as "fat tailed" as the observed data. Given these characteristics are of core importance to the main conclusions of the paper, I think these issues need further discussion.

**G2G modelling**

In their most recent response to reviewers, the authors state that "a direct comparison with observations is not possible" as gridded observations of flow do not exist. This is an oddly naïve statement to make as first, this is obviously the case everywhere in the world, and second, gridded model outputs can be aggregated to represent catchment runoff and compared to observations at selected streamflow gauges; also, if I interpret Figure 1 correctly, isn't this exactly what the authors have done to investigate model bias in estimates of daily 50-year return period (Q50) flows? That is, isn't Figure 1 based on comparing aggregated gridded runoff outputs (or routed flows) with locations where streamflow gauges are available?

My concerns with the current Figure 1 are that no mention is made of what bias-correction method was used (simple delta scaling, quantile scaling, or something more sophisticated? was it applied to the climate projections or directly to the Q50 estimates?); but more importantly, it lumps the potential biases of both the RCM climate projections and the G2G modelling together, so it is not possible to determine which aspect of the modelling chain is causing problems, and thus how the

biases in either (or both) the G2G and RCM modelling should best be addressed. Accordingly, the more useful approach would be to compare the Q50 results derived from SIMOBS results to the Q50 estimates derived from the gauged streamflows as this then reveals the efficacy of the G2G model separately from the efficacy of the RCM simulations, which are already explored in Figs 4 to 7.

I assume that the authors' justification for the adequacy of the G2G modelling relies on the basis of previous published papers. I have thus gone back and searched the literature cited in the paper (noting that the Kay et al. 2018 was missing from the reference list) but I could not find any assessment in the published papers of the G2G model's ability to estimate the extreme floods of most relevance to this paper. On the basis of the model's formulation, I would expect that the G2G model is well suited (if appropriately parameterised) to characterising low and high flow regimes and soil moisture behaviour, but that it would struggle to represent the extreme floods of interest.

Without seeing evidence to the contrary, I would expect that the G2G model would be better suited to water resource applications rather than flood applications as the model structure and parameterisation is focussed on the gross partitioning of rainfall, evapotranspiration, soil moisture accounting, and the redistribution of sub-surface moisture; while these state variables are relevant to antecedent conditions which influence flood behaviour, they are not well suited to characterising the flood response during an extreme event. From a purely information content perspective, 99.98% to 99.99% of the daily data used to inform the G2G parameters relate to non-extreme conditions. Unless special steps are taken to use the ~0.02% of information relevant to extreme flood behaviour (and the model structure and parameterisation is able to take advantage of it), then it is unlikely that such a model is able to adequately represent flood conditions. I would expect such models tend to provide precise information at highly resolved spatial and temporal scales, but the estimates are likely to be biased and inaccurate, particularly when the model has been configured to provide estimates at regional/national scales rather than catchment. I do understand the value and trade-offs involved in developing national-scale models as opposed to catchment-specific models, but I also think we need to be transparent about how the performance of such modelling schemes.

Of course I may not be right in this instance as my expectations are based on my own experience of using a range of conceptual and "physically based" models, and not of this G2G model. However, my difficulty here is that I could not find any evidence in the current manuscript, or in previously published papers, that the G2G model is able to provide reasonable estimates of the Q50 flood. My above recommendation for comparing the Q50 estimates derived using the SIMOBS inputs would provide clear evidence as to how well the model is performing at the national scale of interest, and these insights can be used to add defensibility, or caveats, to the conclusions drawn.

Rory Nathan
University of Melbourne

---

## Author Response (AR2)

**Response to Reviewers and Editor**

Responses to the reviewers are given in blue following each point. Line numbers correspond to the first revision, not the new version.

**Reviewer 1**

**General Comments**

The motivation of the paper to examine the spatial and temporal coherence of flood risks is well justified and the authors' approach to examining this topic is efficient and effective. However, the justification for using raw climate data for examining floods resulting from extreme climates is not yet satisfactorily justified, but I believe an additional validation step would allow the authors to retain the approach they have already adopted. I find the manuscript to be overall well written and figures are mostly well presented (some improvements are needed for Figs 3, 4, 7, and 8 and Figure 5 needs to be revised). A number of corrections and clarifications are necessary to ensure that: consistent terminology is used; the language considers the global readership of this journal; descriptions are precise and objective; figures are easily interpreted; and referencing is accurate.

**Major comment**

I concur with the authors that the use of bias correcting in an attempt to capture rainfall extremes relevant to floods is difficult and introduces considerable uncertainty and the justification here is to avoid the introduced uncertainty and instead examine the differences in floods resulting from modelled historical and future climates. However, the justifications currently provided are inadequate and at times illogical. The authors state that "due to the focus on [sic] the present work on extremes rather than the whole regime in general, bias correction is not applied here."

I'm afraid this justification is unsatisfactory. I'm certain the authors are knowledgeable of the fact that regional climate models are aimed at resolving large scale spatial and temporal variability, namely, what is referred to here as "the whole regime in general". A study that was aimed at large scale changes would therefore be well justified in using the raw RCM outputs. In contrast, extreme precipitation events are typically driven by synoptic scale events, which would justify the use of bias-corrected projections. Furthermore, flooding is sensitive to the spatial and temporal distribution of storms at even finer scales and have the added complexity of local factors that influence the flood response other than climate. The focus on extreme events is therefore a justification for introducing uncertainty to address the increased complexity of flood responses. Lines 65-66 therefore needs to be removed.

Lines 65-66 have been removed. Instead the following has been added at the end of Section 3.1

An investigation was undertaken to identify whether bias correction should be used in this paper. G2G outputs based on the UKCP18 RCM ensemble members was compared to daily mean flow (data available from the NRFA). The 50-year event (annual exceedance probability of 2%) was calculated for the station and the relevant gridsquare it lies in. Figure 1 shows that across most of Great Britain where the mode was run, bias correction led to a fairly constant underestimation of the 50-year event compared to those from observations. Although the results without bias correction are more variable with some stations showing a large overestimate, they have a better mean bias when calculated nationally, which was felt to be important when looking on a national scale. This was also computed for the 2-year flood with very similar results. Due to this, it was decided that bias correction would not be applied in this paper.

[Figure]

**FIGURE 1 COMPARISION OF DIFFERENCE BETWEEN (A) WITH AND (B) WITHOUT BIAS CORRECTION AVERAGED OVER ALL 12 ENSEMBLE MEMBERS. COLOUR INDICATED THE CHANGE BETWEEN THE ESTIMATE FOR THE 50-YEAR RETURN PERIOD PEAK FLOW (BASED ON GAUGED DAILY FLOW). POSITIVE VALUES INDICATE THAT MODELLED DATA HAS A LARGER VALUE OF Q50.**

In their response letter, the authors make the case that they have kept the biases constant by comparing modelled historical and modelled projected floods and by capitalising on the dynamic nature of the GCMs. (however, it then follows that the same justification would not rule out the comparison of bias-corrected data – here too, the biases would be consistent between modelled historical and projected climates. Furthermore, bias corrected rainfalls are not constrained to the observation based datasets, as it appears to be suggested in the authors' response– both statistical and dynamical downscaling approaches allow for estimates outside the range of the observed records.)

This justification is sufficient and acceptable conditional upon the authors adopting the following suggestions, which should be easy to implement:

1. Repeat the analysis using observational data over the same period as the historical modelled data to quantify the degree to which the spatial and temporal coherence of flooding is approximated and represented in the baseline case. This would then provide evidence of the fidelity of the baseline to which the relative changes in the number of widespread events could be estimated using the approach that has already been presented. Currently, the presentation of results dependent only on modelled climate data makes it easy for a reader to suppose that the results reflect similarities in catchment characteristics combined with the lack of spatial specificity in the modelled climate date. Having an observation-based analysis would provide context. Presenting these observed results in the SI would be adequate

2. Present the findings as relative changes in the number and spatial scale of the widespread events (more details of this in reference to figure 3 below)

> We thank the reviewers in their continued interest in this point. Unfortunately, gridded observations of flow do not exist nationally for the United Kingdom, and so a direct comparison with with observations is not possible. However, we do now include a comparison with some "observation-driven" simulations (SIMOBS), which have been discussed and analysed previously in Kay (2022). These observation-driven simulations make use of observed gridded rainfall, gridded potential evapotranspiration and gridded daily observed temperatures. We add panels or bars to Figures 3, 4, 5, and 6 which show the values for the SIMOBS run (which only exists for the baseline timeslice).

The following is added to the methods section:

In isolation it is difficult to say whether these results are realistic compared to observations. However, gridded river flow observations are not available for Great Britain nationally. Therefore, in addition to the UKCP18-driven G2G output, this paper also presents data from a set of "observation-based simulations" as used in Kay (2022) as a step towards comparing modelled and observed extreme flow. This run still uses Grid-to-Grid but is driven using observed inputs: CEH-GEAR daily gridded precipitation (Tanguy et al., 2016), monthly short grass potential evapotranspiration (40km resolution) from MORECS (Hough and Jones, 1997), and daily 1km minimum and maximum daily temperatures (Met Office et al., 2019). Precipitation was subdivided uniformly through the day, and temperature varied sinusoidally between the extremes. In this paper, this will be referred to as the SIMOBS run.

The following is added at line 194 in reference to Fig 3

Fig 3 shows that the event areas are fairly consistent between the RCM-driven runs and the SIMOBS run, with a slight bias in the RCM-driven runs to larger events with lower return periods. All the RCM-driven runs show a slightly flatter distribution of return periods in the 2050-2080 time-slice.

The previous version of Figure 3 (with some minor formatting changes) has been moved to Supplementary Material Figure 1.

The following is added at line 211 in reference to Fig 4

In comparison, the SIMOBS run shows a more equal distribution of events across the seasons, though it still remains within the variability of the seasonal totals for the RCM-based baseline outputs except in autumn (SON).

The following is added to line 225 in reference to Fig 5.

The SIMOBS run appears to generate shorter events on average compared to the RCM-driven runs, suggesting a slightly stronger temporal autocorrelation in the effects of the use of UKCP18 input data. The return periods (as seen in Figure 3) are broadly similar in distribution.

The following is added to line 232 in reference to Fig 6.

The SIMOBS run shows a broadly similar distribution to the baseline (1980-2010) timeslice, although the variability and reduced smoothness appears to be increased, due to the much smaller number of events from that single run (~500 compared to ~7000 from all 12 RCM ensemble members).

**Specific comments (line numbers reference manuscript 2)**

Acronyms are often used before they are defined – the ones I noticed were: NRFA, RCM (this shouldn't be assumed knowledge since GCM is previously defined), PoE. There may be others.

       All abbreviations spelled out on first usage.

It would be helpful to be clarify the terminology used to describe seasons - this is primarily for your readers in the southern hemisphere, please don't neglect us. Please consider including "boreal" when seasons are referenced. Alternatively, include the months in brackets after each season. It may seem superfluous for a northern hemisphere native, but it makes interpreting the results so much easier for those in the south.

       Months are referenced when seasons are mentioned, e.g. spring (March-May), summer (June-Aug), autumn (Sep-Nov) and winter (Dec-Feb).

The time slices are introduced as a "baseline" and "future", but the former is later referred to as "present". Please keep the terminology consistent, particularly since "present" is inaccurate when referring to a time period covering 1980-2010.

       We agree with this comment. All references to "present" are swapped with "baseline".

L 90: The term "percentiles" in reference to floods is used to describe distributions. The term that should be used here is frequencies, not percentiles.

*We change this to "frequencies".*

L95: similarly, the notation Qx is widely used in hydrology to denote the x percentile of flow rather than a flood frequency. Please keep the terminology you have chosen consistent by referring to the 1 in 5 and 1 in 10 year probability event as POT 0.2 and POT 0.1 respectively.

Alternatively, use the term Average Recurrence Interval of 5/10 years (ARI5 and ARI10).

*We will use POT0.2 and POT0.1 throughout instead of Q5 and Q10.*

L99: This is ambiguous as absolute values of flow are in fact being used as thresholds, and these thresholds are dependent on the distribution of the data: the definition of the thresholds are just location dependent. I suggest phrasing this as:

Note that the thresholds are not based on universally applied fixed values of flow magnitude but are instead dependent on thresholds defined by empirical flood frequencies.

*Thank you for this suggestion. We replace the sentence with the one offered.*

L150: I had to read this several times and I'm still unclear. The text implies that $\bar{\chi}$ applies to large flows and conversely that $\chi$ applies to flows of all magnitudes (which I do not believe is the intended message). I suggest the following (but I'm unsure whether I've interpreted the text correctly. Please revise as necessary)

$\chi$ describes the level of asymptotic dependence; if $\chi^1 0$ then the variables are asymptotically dependent. A value of $\chi=0$ represents asymptotic independence. Asymptotically independence is also represented by a value of $\bar{\chi}^1 1$. A value of $\bar{\chi}=1$ represents flows that are dependent but nor asymptotic.

*We replace the existing paragraph at line 150 with: $\chi$ describes the level of asymptotic dependence; if $\chi > 0$ then the variables are asymptotically dependent, and $\bar{\chi} = 1$ automatically. But if $\chi = 0$, they are asymptotically independent. In this case, $\bar{\chi}$ describes the dependence for large but not asymptotic values of flow. $\bar{\chi}$ close to 1 indicates the variables are highly dependent except at the asymptotic limit.*

L185: please justify why these four events are selected. Why "four of" and why not "the four largest"? Are they selected to demonstrate that most events are spatially contiguous? (As an aside to the major comment above, is the spatial coherence an attribute derived from the modelled climate data, or is the pattern present in observed data?)

*We correct line 176 to say "Fig 2 shows the four events with the widest spatial extent…" Due to a lack of gridded observed flow data available for the United Kingdom, it is not possible to compare these events with observed events.*

Figure 2: If I'm understanding this correctly, this figure shows the degree of spatial inundation with the colour scale showing the equivalent severity of the flood for events identified using the POT2 threshold, equivalent to a return period of 0.5. If this is the case, why are more frequent events with return periods of 0.2 to 0.5 shown? Perhaps they are not, but the light yellow scale is difficult to distinguish between greater or less than 0.5. I suspect it is simply a case of the legend needing to be updated to show grey between 0.2 and 0.5. Also, the text size of the labels on this and the next figure are rather disproportionately large. Please reduce the text size.

*The reviewers are correct – the light yellow should be coloured grey, no points on the map are actually coloured using that shade. We also correct the label sizes.*

Figure 3: Comparing the changes between the baseline and future is not easy with the way this figure is presented – there are a lot of vertical bars in different ensemble members to compare and lining up the number of events for different return periods is challenging, while the colour selection implies that A and B are showing a different variable to C and D. I strongly suggest combining the information from A and C, and B and D by showing these results as the difference between time slices. If there is a very good reason to not do this, please at the very least change the colours to paired colours: e.g. light blue for A, blue for C, light green for B

and green for D, so that light colours correspond to the baseline, darker colours to the future, blues for area, and greens for return period.

To compare with an additional run using "observation-driven simulations (SIMOBS)" as discussed above, Figure 3 has been moved to the supplementary material, and replaced with a figure that compares the SIMOBS with a weighted average of the ensemble members for the baseline timeslice and the future timeslice (weighted by number of events extracted), which scales the number of events to be comparable to the SIMOBS run. This combines panels A/C and panels B/D, using shading for the ensemble average and two outlines for present and future.

L 203-2007: The choice of using the word "may" makes the sentence sound speculative. These sentences could be revised to reflect the certainty of the results. Suggest the following or similar:

However, the increase in widespread events is confined to the boreal autumn (SON) and winter (DJF) with a decrease in events between March and August. The decrease in future boreal spring and summer events could be due to overall projections of drier summers (Murphy et al., 2019), or could result from a spatial contraction of summer floods in the future, which have historically resulted from short-duration, high intensity storms.

We appreciate the reviewer's suggestion and agree that it is more appropriate. The change suggested is applied.

L 212: It's unclear what the term "methods" is in reference to. Is it the method of climate modelling or is this meant to mean differences in flood responses to different storm types?

Methods should be "models". The sentence now reads "Thus future work could build on Kay (2022) and look more specifically at differences between RCMs and CPMs which could explain these patterns."

Figure 4: The left figure is superfluous. Please remove this.

Agreed and removed.

L227-230: Describing the changes in reference to changes in return periods is unconventional and is probably due to the way the figure axes are configured (see comment on Figure 5). Please change this description to one that describes the shift in event duration with respect to frequency.

We adjust this sentence to read "… somewhat correlated, in that the events with the highest return periods are very unlikely to be of short duration.

Figure 5: is there actually a heavier tail in the future in SON? (L223) Also, the return period must be plotted on the x axis, as the duration of the event is dependent on the return period not the other way around. This isn't just a formatting choice – the reader is forced to attempt to transpose figure 5 as the convention in referring to tails is the distribution along the horizontal (with a few exceptions). In addition, there isn't a good reason to present Figures 5 and 6 in a way that is disjointed. The reason given in response to the previous reviewer of duration having a smaller range than return period (and likewise in Figure 6 the return period having a smaller range than area) is disputable: one could easily represent the return period in log10 years and have an effective scale of 0-4 (just making a point – I'm not suggesting that return periods be presented in this way).

Figure 5 has been flipped to be consistent with the other figures in this paper, addressing the problem you present.

Figure 7: labels a, b, c, d are missing from Figure 7 (b is referenced in the text). The caption needs to describe what is in each of the four figures (i.e. different time slices and different measures). This figure is also inconsistent with previous figures that have shown the baseline time slice on the top row and the future on the bottom (this can be fixed and the legends could simply be placed horizontally under the respective columns). Using a different color scheme for each metric would aid in interpreting the different scale of results.

Fixed as per suggestions.

Figure 8: as per Figure 7 regarding layout and caption. Only one legend is needed.

**Minor comments**

Grammar and style

L66: "of the present work"

Removed as relevant to bias correction work elsewhere

L175: "shown" instead of "show"

Fixed

L231-233: missing "there"; "matched" not "matches"; "dynamics" is not an appropriate adjective here. I think what is meant is: but there is variability amongst the ensemble members in the relationship between event duration and frequency.

Fixed: replaced "dynamics" with "patterns".

L237: "is not surprising": please replace this with objective language e.g. statistically plausible

Fixed

Referencing: Following on from a previous reviewer's comment, please check the accuracy of all references. One example is that of Tawn et al. 2018: it is referenced as 2019 in the text and the list of authors in the reference list is incorrect. The dates for Towe are also reference incorrectly in text. There may be others, so it may be prudent to check the referencing system.

This reference has been fixed, and re-proofread. There are two references here which have been confused, and so this has been corrected: Tawn et al. (Spatial Statistics, 2018) and Towe et al., (J AGRIC BIOL ENVIR S, 2019).

**Reviewer 2**

**General**

As stated previously I do think the strength of the paper is its use of a climate model ensemble to simulate areal flood events, where the joint interaction between the factors that cause floods over a range of temporal and spatial scales is implicitly accommodated by the use of a gridded daily continuous simulation model. I have gone through the various comments and responses, and overall I appreciate and agree with the changes made by the authors.

I appreciate that the authors have undertaken a computationally demanding set of simulations – the data sets and modelling framework are impressive – but I regret that I am left with concerns about two major points which I raised previously, namely 1) the need to compare baseline modelling with observations, and 2) the defensibility of the probability of exceedance estimates. The authors' response to the former point focussed on the problems with bias correction without addressing the more fundamental concern around the need to demonstrate how well the modelled baseline frequency curves of areal rainfalls or floods conform to observations. The response to the latter point (the method of calculating probabilities of exceedance) does not provide additional confidence in the validity of the results and some reconciliation of the different estimates is needed.

This is discussed in more depth for Reviewer 1's very similar point. In brief, no gridded flow observations exist for the UK, but we compare the results to "observation-driven simulations" as used in Kay (2022). With regards to bias correction, we include a brief discussion where it was shown that, nationally, bias correction at locations with stations systematically underestimated return periods across the country compared to those observed, and without bias correction this systematic underestimation was lessened. Thirdly, the distribution has been swapped with a standard Generalised Pareto distribution using a simple scaling factor to convert from a per-exceedance to an annual probability.

I provide more commentary on these two points below.

**Baseline Evaluation**

I agree with the authors' caveats about the dangers of bias correction, but my main point in this regard was the need to provide evidence that the frequency distribution of areal event extremes derived from the UKCP18 data compare reasonably well with observations. My earlier comments expand on this point a little, but as far as I can tell the authors' justification for not examining such evidence is because they adopted flow thresholds to yield a specified number of flood exceedances over a given period (and over a given area). This approach does not demonstrate how well the results generated over the baseline period relate to real-world conditions, it is merely a device to extract the number of events relevant to the exceedances of interest. What is missing here is a comparison of the selected thresholds with what has been observed over some suitable baseline period.

In other words, the selection of thresholds to yield a defined number of flood occurrences provides no information on how well the magnitude-frequency relationship of the flood regime is preserved (ie how well the cumulative density function governing the extremes matches reality), and this is a particular problem as the modelled floods are fitted to a Pareto distribution and the results are reported on in terms of absolute shifts in return periods (ie in terms of a shift in the magnitude-frequency relationship).

The results derived by fitting a Pareto distribution to the modelled events are very likely to be impacted by various forms of bias, and varying the thresholds to achieve the required number of exceedances does not avoid or obviate the need to undertake bias correction, it just ignores the problem. The degree of difference between modelled and observed flood frequency curves for some representative locations would add considerable weight to the conclusions; without demonstrating this, the conclusions would need to state clearly that the adopted methodology yields results that have not been evaluated against observations over the baseline period, and thus are rather more speculative than currently framed.

> See our discussion on using SIMOBS in our comments to reviewer 1 for more detail. There is a lot of new content throughout.

**Probabilities of Exceedance**

I also think we should be troubled by the difference in results obtained using the two procedures to estimate the probabilities of exceedance (ie the one based on deriving annualised exceedances from a fitted GPA distribution with an assumed spatial dependence, and the method as described in the paper).

The authors have stated that they wish to retain the method as outlined in the paper as it is being used in another manuscript and they wish to "minimise confusion". While I appreciate the convenience of this decision, if the method cannot be shown to be correct then I see little point in being consistent.

The problem is that we have two methods (applied with different levels of rigour) that have two markedly different risk implications: I suggest that the subsequent check made with the assumed degree of spatial dependence yields results that are consistent with expectations, whereas the results presented in the paper are not. I think this a problematic outcome.

Accordingly, I think additional investigation is required to reconcile these different probability of exceedance estimates. The differences in results are too great and too surprising to ignore, and some diagnostic checks should be devised to check whether the inconsistencies are due to computational errors or unsupportable assumptions in one or other of the approaches.

Rory Nathan University of Melbourne

> We thank the reviewer for this considered response. To reduce the complexity of both the computation and the discussion, and since we only care about threshold exceedances, we restrict this paper to a standard Generalised Pareto distribution as suggested in the previous review. Since the probabilities are still "per exceedance", we convert to annual probabilities using a simple scaling factor (# events extracted ÷ #

events per year). This reduces the number of events with a return period exceeding 1000 years to a more plausible level.

**Reviewer 3**

The paper investigates the change to wide-spread flooding in terms of number of occurrences duration and extend between current and future climate using an ensemble regional climate model. This is a very relevant study showing novel results. However, revisions are required as per the comments below.

**Main comments**

The novelty and aim of the paper is not clear after reading the introduction. In the introduction the relevance of the work is clearly discussed, however no clear research gaps are mentioned. Also, at the end of the introduction (L46-50) a short summery of the methods rather than the aim of the paper is provided.

Some clarifications are needed in the data and methods sections (see specific points below). Especially, the statement that bias correction of precipitation from climate models is not required when focusing on extremes needs explanation. Furthermore, I think if the sensitivity of the assumptions (event magnitude threshold, extent threshold, and maximum event duration) in section 3.2 should be discussed separately from the methods as the current structure makes it hard to follow the methodology. And the manuscript would really benefit from a discussion on how these assumptions affect the final results (rather than the number of events for a single ensemble).

The conclusion section also contains discussion of the results as well as some limitations. For clarity, I recommend splitting the conclusions and discussion by making a separate section on limitations and moving the discussion of the results in relation to other papers to the "results and discussion" section.

Firstly, we add at line 50 "Often flooding is considered on a site-by-site or regionally summarised fashion, particularly when looking into projections of the future. This paper hopes to show the benefits of considering widespread flooding events over a large area using gridded, rather than catchment-based hydrological modelling to expand our knowledge of the extent of possible flooding events in the UK. Comparing UKCP18-driven model runs to those driven by observed rainfall and temperature will give confidence to the use of these event sets in future analysis." To the introduction to highlight our aims and the novelty of the work in the rest of the paper.

The discussion of bias correction is now greatly lengthened, see out comments to Reviewer 1 for more detail.

The discussion and conclusion sections have been restructured to remove new discussion points from the conclusion, and move limitations to the discussion section as well. Discussion points which do not specifically count as results have also been moved to the discussion section.

**Specific comments**

L12: "…, allowing events to last up to 14 days" seems to suggest that the 14 days are a consequence of the event definition given before, but I don't see how this is the case. Can you clarify?

Changed to "with a maximum duration of 14 days" for clarity.

L36: I suggest to leave out "driving" in "driven by large ensembles of driving data from climate models" as it seems double?

Removed the word "driving"

L37: Can you provide more information about what "predominantly stochastic event-based models are"?

> We replace this with "…, or through Monte Carlo methods simulating boundary conditions to feed into an event based model (PQRUT) (Filipova et al., 2019)."

L42: Replace "focusing on the United states" by e.g. "for the United States", as using "focusing" twice in this sentence is confusing

> Replaced with "for the United States"

L54: This is the first time UKCP18 is explained while it has already been used in the abstract and introduction. Please explain at the first use.

> All abbreviations are now explained at first usage.

L66: The authors seem to state that bias correction of precipitation from climate models is not required when focusing on extremes? I would disagree with this and would like to see more better argued why no bias correction is required.

> As discussed above for Reviewer 1, a small example is given showing that bias correction leads to a systematic underestimation of the 2- and 50-year events observed at stations.

Section 2.1: Why was RCP8.5 selected? It would be good to mention this choice also in the abstract.

> RCP8.5 was the only scenario available from the UKCP18 gridded rainfall datasets at the required resolution – there was no choice available to be made. We add "- the only available scenario in the 12km grids (Riahi et al., 2011)" to line 59.

Section 3.1 It would be good to add which version of the model is used. Whether the model has been calibrated for this study or a previously calibrated version has been used.

> We add the following to line 80: "This paper uses the same version of Grid-to-Grid as used in Kay et al., 2018 and Kay 2021, since it uses the same driving data."

L105: "this was considered equivalent …." This subsentence is confusing to me. Please consider leaving it out or clarify its meaning.

> This subsentence is replaced with "(denoted 'inundated')".

Table 1: "PoE" is not explained. Furthermore, it would be clearer if consistent naming of thresholds were used in the "exceedances" column and in the text (e.g. POT2 or 2/yr). Also note the typo in "exceedances".

> Another proofread has been undertaken, and all abbreviations are spelled out on first occurrence.

L119: I don't understand why based on minimum-threshold, flood extents that are smaller than this threshold are "retained". I would expect these are excluded. Could you please clarify?

> Sentence changed to "The 0.1% inundation coverage was selected to ensure that small, very extreme events were not excluded."

L154: Why were 60 peaks selected and how does this relate to earlier event magnitude threshold? Also, from which ensemble or for all ensembles? Please clarify.

> 60 peaks matched an average of two events per year in a 30-year timeslice. Each ensemble member was fitted separately. We have added "the top 60 independent peaks *in each ensemble member and timeslice* were found…".

L159: How is the "daily exceedance probability" calculated from 60 peaks? I'm used to converting these probabilities to annual exceedance probabilities directly based on the average number of peak events per year, which you seem to refer to as an "alternative approach" (L171).

> As discussed for Reviewer 1, in this paper, we have switched to just using a standard Generalised Pareto distribution, rather than the previously stated mixed distribution, and as you have mentioned, swapped to converting to annual probabilities using a simple scaling factor (# events extracted ÷ # events per year).

L174: Could you provide more context to the sentence "which might potentially align with discussion of the frequency of 100-year events in the UK"? What is this discussion about?

This sentence has been removed to improve clarity.

Section 4: In Figure 3, 5, and 6 event return periods are shown (if I understand correctly). It is however unclear to me how these are calculated? As show in Figure 2 the return period varies spatially, and it is not clear how a single return period is calculated.

The sentence "In the rest of this section, return periods reported in the text and figures are the maximum return period observed (across space and time) within a single event." has been added to line 186.

L234: I assume AMAX is "annual maxima"? This has not been explained before.

All abbreviations are spelled out on first occurrence.

L237: What do you mean by "change in flow"? Is that change in peak event magnitude?

Changed "flow" to "event peak flow magnitude".

L247: Can you explain what the value of 120 km is based on? In the figure it seems there are still points whit significant asymptotically dependence up to 250 km.

In this sentence, we add "… have a limit at most location pairs of around 120km…"

L271: I would be very useful to understand how sensitive the change in number of events is to the selection of thresholds in section 3.2. And how significant it is given the differences between ensemble members.

We add this sentence to line 114. " Very similar patterns of events extracted (not different at a statistically significant level) were observed for all of the ensemble members."

L299: Can you clarify what you mean by surface water flooding as opposed to fluvial flooding?

Surface water flooding refers to flooding caused by means other than the overtopping of a river or water body. For example, high-intensity rain storms in paved urban areas can result in surface-water flooding if drainage is insufficient.

**Editor Comments**

The overall comments from the reviewers highlighted to us the need to go back and do further work to justify the use of a) the modelled UKCP18 data and b) not to use bias correction. We also reduced the complexity of the probability distribution used to a simple GPa distribution with a basic scaling factor to get annual probabilities. This leads to a better link up to other work in this field. This all led to two new sections of work in the paper (explained in our comments to Reviewer 1) and we feel it has benefited the paper over all. We hope that this addresses all the remaining concerns.

---

## Author Response (AR3)

**Response to reviewer comments on "Widespread flooding dynamics changing under climate change: characterising floods using UKCP18"**

The authors would like to thank the 3 reviewers and the Editor for their further comments, and their perseverance in helping us improve the manuscript. We describe below how we have addressed each comment, and sincerely hope that the reviewers and Editor now consider that we have adequately dealt with all of their comments and that the manuscript is now suitable for publication.

**Editor:**
From my reading of the manuscript as an Editor I concur with reviewers #1 and #3. I encourage the authors to revisit their approach along the lines of the suggestions of reviewers #1 and #3.
Please see our response to each reviewer comment below.

I find it disappointing that the authors did not address my own comments. I still think that an article published in an international journal such as HESS must be of interest to readers outside the authors' country, e.g. by reporting methodological advances. I am sorry to say that the authors did nothing to emphasise the method over the UK-centric implications. They conclude that the number of widespread events in the UK and their seasonality will increase. Why would this be of interest to anyone outside the UK? Without an attempt to address this concern, the paper will unlikely become acceptable.
We have now made a number of changes, with the aim of making the manuscript of greater interest internationally and less UK-centric. These include

- Changing the title to make it more general.
- Adding more about the method to the abstract, as well as emphasising that the methods/analyses could be applied elsewhere.
- Edits to the Introduction to make it more general, and again emphasise that the methods/analyses could be applied elsewhere.
- Separating the Discussion into a part relating to the GB results and a part relating to the methodology, with the latter now including discussion of what particular aspects of the method may need to be specifically considered for applications in other regions.
- The Conclusions now stating "While the focus here was GB, the methods and analyses described could be applied to other regions with hydrological models and climate projections of appropriate resolution. The most suitable definition for a 'widespread flood event' may vary for other countries/regions."

We hope that these edits now make the manuscript of much greater interest internationally.

**Reviewer 1:**
**Overall comments and recommendations**
The authors have given this revision of the manuscript considerable thought and effort, and the additional material helps to highlight both the challenges and value of the work undertaken. I raised two major concerns previously, namely 1) the need to assess the efficacy of the model simulations against suitable reference data, and 2) the manner in which the probabilities of exceedance were estimated. I am comfortable with the changes made to address the latter concern, so my comments below are focused on the efficacy of the adopted modelling chain. I sat on my draft comments for a week before finalising as I would like to see the paper published and I appreciate that the authors will be getting frustrated with this review process; however, I am struggling to agree with the author's interpretation of the new evidence presented and to date only the RCM projections and not the G2G model itself has been assessed.
Thank you for your continuing efforts to help improve our manuscript. We hope that the substantial edits and additions now made, described below, are sufficient to justify acceptance/publication.

The key contribution of the paper rests on characterising the possible changes to areal flood behaviour under climate change, where the adopted method implicitly accounts for the joint interactions between soil moisture, rainfall, and the non-linear influence of increasing catchment scale. The work leverages a large body of prior work undertaken to provide the adopted regional climate model (RCM) ensembles and G2G (hydrologic) modelling; while the use of this inherited work is of great benefit, it also brings with some specific challenges relevant to the stated research objective.

It is true that the work reported in this manuscript relies upon a large amount of preceding work, which has both advantages and disadvantages. It also meant that, in the interests of brevity, we were trying not to repeat too much from elsewhere – a difficult balance, that we got a bit wrong but have hopefully now improved!

My interpretation of the new information provided in the last revision is that the RCM ensembles do a poor job of simulating the temporal and spatial correlations of the key processes of interest, and this undermines the core conclusions of the paper as currently framed. It also raises questions about the defensibility of the G2G simulations. While it is perhaps unreasonable to expect that the authors somehow "fix" deficiencies in the prior modelling effort that they have inherited, I do think it important that any shortcomings in the modelling chain are explored and used to qualify the conclusions made and to highlight areas for future research.

The differences in flood characteristics between the baseline climate model-driven run and observation-driven run have now been described more fully in the Results (Section 4), and a paragraph has been added to the Discussion (end of Section 5.1) discussing the differences and the possible reasons for the them, and areas for future work. The need to interpret future changes "in the context of any differences in event characteristics between the baseline climate projection-driven model runs and an observation-driven model run" has also been added to the Abstract and Conclusions (Section 6 para' 2).

It appears that my assessment of the new information differs from the stated views of the authors, and this may be because I am misinterpreting information presented in the manuscript. So, either my concerns point to the need for the authors to better clarify and justify their reasoning, or else to perhaps revise their views based on the comments below.

I think the contribution of the paper would be greatly enhanced if the authors:

> 1) replace the current Figure 1b with an equivalent figure based on G2G results forced by "observation-based simulations" (rather than by the current 12-member RCM ensemble), and replace Figure 1a with a simple scatter plot of observed vs simulated Q50; the former plot provides insights about spatial biases related to regional hydroclimatic differences, and the latter plot about the nature of any overall model biases.
>
> 2) adopt a more critical approach to reviewing the implications of the results from the proposed G2G assessment as described in the previous point, and also from the RCM ensemble assessment already undertaken and currently summarised in Figures 4, 6 and 7.

The effort involved in undertaking 1) should be modest as the necessary hydrologic simulations have already been undertaken, but if this is not possible for some reason, then the manner in which the discussion and conclusions are written could be revised to better highlight, or refute, the perceived limitations in the hydroclimatic projections on which the main conclusions are based.

1) The suggested replacement for Figure 1b (a map showing the flood performance of the SIMOBS run) is available elsewhere (Kay 2022) - we have now added a paragraph describing these results and some older work that shows similar patterns of performance (Section 3.1 para' 3) (unfortunately an oversight meant that the older work was not cited in this manuscript before). The recent work also includes maps of performance with and without a simple bias correction of

precipitation – information on this has been added (Section 2.2 para' 2), and Fig 1a replaced with the suggested scatter plot (Section 3.1).

2) As described above, a paragraph has been added to the Discussion about the differences between the SIMOBS and baseline RCM-driven results, the possible reasons for the them, and areas for future work. The need to interpret future changes "in the context of any differences in event characteristics between the baseline climate projection-driven model runs and an observation-driven model run" has also been added to the Abstract and Conclusions (Section 6 para' 2).

Further rationale for these recommendations is provided below.

**RCM projections**

Comparing the 1981-2010 flows derived from the RCM ensemble with those from the "observation-driven" inputs provides a very effective evaluation of the modelled climate over the 1981-2010 period. It is more valuable than comparing any individual climate variable alone (e.g. rainfall) as it implicitly allows for the joint distribution of climatic factors that influence floods, so this is a very useful addition to the paper.

However, at present the authors conclude that the RCM simulations of event areas are "fairly consistent" with those derived using SIMOBS, and that there is "slight bias" in the distributions of event severities. I find it difficult to accept these conclusions as stated because:

1) the differences in results between the SIMOBS and the mean RCM ensemble simulations as shown in Fig 4 are appreciably larger than the modelled differences between the climatic baseline and future conditions, and

2) the SIMOBS results appear to lie outside the maximum and minimum range of individual ensemble RCM results over the period 1981-2010 as deduced from a comparison between Fig 4 of the MS and Figure 1 of Supplementary material.

The differences in results would be better illustrated by including 'error bars' in Fig 4 that show the min/max spread of ensemble results for the 1980-2010 period, or else illustrating these differences in some other more easily comprehended fashion. It would also appear that the RCM simulations greatly overestimate the duration of the events (Fig 6), where it is observed that the differences between the RCM ensembles and the SIMOBS results are considerably greater than the projected differences due to global warming. The simulation of flood extents (Fig 7) appear more reasonable (but there is still an order of magnitude difference between simulated and observed maxima), and I agree with the interpretations of the authors regarding the adequacy of the seasonality of the flood regime as shown in Fig 5.

Error bars have been added to Fig 4, in a similar way to Fig 5.

It is difficult to assess how well the heatmaps for SIMOBS vs baseline RCM-driven compare (Figs 6, 7), due to the much more limited number of events available from the SIMOBS run than the pooled baseline RCM-driven runs. This is particularly the case when considering combinations of properties (as for the heatmaps, rather than the preceding histograms), and particularly for more extreme combinations (as is pointed out in the Discussion, Section 5.1 para' 1).

As described above, the differences in flood characteristics between the baseline climate model-driven run and SIMOBS have now been described more fully in the Results (Section 4) and a paragraph has been added to the Discussion (end of Section 5.1).

In short, the results shown in Figures 4, 6 and 7 appear to suggest that the RCM climate ensembles overestimate the spatial dependence of rainfall events and over-estimate the serial dependence in the correlation structure of rainfall extremes, and further that the distribution of modelled areal maxima is perhaps not as "fat tailed" as the observed data. Given these characteristics are of core importance to the main conclusions of the paper, I think these issues need further discussion.

As described above, a paragraph has been added to the Discussion (end of Section 5.1) about the differences between the SIMOBS and baseline RCM-driven results, the possible reasons for the them, and areas for future work.

**G2G modelling**
In their most recent response to reviewers, the authors state that "a direct comparison with observations is not possible" as gridded observations of flow do not exist. This is an oddly naïve statement to make as first, this is obviously the case everywhere in the world, and second, gridded model outputs can be aggregated to represent catchment runoff and compared to observations at selected streamflow gauges; also, if I interpret Figure 1 correctly, isn't this exactly what the authors have done to investigate model bias in estimates of daily 50-year return period (Q50) flows? That is, isn't Figure 1 based on comparing aggregated gridded runoff outputs (or routed flows) with locations where streamflow gauges are available?

The outputs from the G2G model are in fact river flow (routed runoff), not grid-cell runoff – this has been clarified (Section 3.1 para' 1). Indeed, previous work has evaluated flow simulations from G2G against gauged flows for locations across GB, and some more recent work on this has now been described/cited (Section 3.1 para' 3). The point we were trying to make when stating "a direct comparison with observations is not possible" was relating to features like flood extent and coherence, which cannot be properly investigated using gauged flow data for a greatly more limited set of river locations than are available from our gridded model outputs.

My concerns with the current Figure 1 are that no mention is made of what bias-correction method was used (simple delta scaling, quantile scaling, or something more sophisticated? was it applied to the climate projections or directly to the Q50 estimates?); but more importantly, it lumps the potential biases of both the RCM climate projections and the G2G modelling together, so it is not possible to determine which aspect of the modelling chain is causing problems, and thus how the biases in either (or both) the G2G and RCM modelling should best be addressed. Accordingly, the more useful approach would be to compare the Q50 results derived from SIMOBS results to the Q50 estimates derived from the gauged streamflows as this then reveals the efficacy of the G2G model separately from the efficacy of the RCM simulations, which are already explored in Figs 4 to 7.

Reference has been added to previous work showing SIMOBS flood peak performance vs gauged data (Section 3.1 para' 3). A discussion of bias correction has been added (Section 2.2 para' 2), and Fig 1 has been replaced with a scatter plot (as suggested above).

I assume that the authors' justification for the adequacy of the G2G modelling relies on the basis of previous published papers. I have thus gone back and searched the literature cited in the paper (noting that the Kay et al. 2018 was missing from the reference list) but I could not find any assessment in the published papers of the G2G model's ability to estimate the extreme floods of most relevance to this paper. On the basis of the model's formulation, I would expect that the G2G model is well suited (if appropriately parameterised) to characterising low and high flow regimes and soil moisture behaviour, but that it would struggle to represent the extreme floods of interest. Without seeing evidence to the contrary, I would expect that the G2G model would be better suited to water resource applications rather than flood applications as the model structure and parameterisation is focussed on the gross partitioning of rainfall, evapotranspiration, soil moisture accounting, and the redistribution of sub-surface moisture; while these state variables are relevant to antecedent conditions which influence flood behaviour, they are not well suited to characterising the flood response during an extreme event. From a purely information content perspective, 99.98% to 99.99% of the daily data used to inform the G2G parameters relate to non-extreme conditions. Unless special steps are taken to use the ~0.02% of information relevant to extreme flood behaviour (and the model structure and parameterisation is able to take advantage of it), then it is unlikely that such a model is able to adequately represent flood conditions. I would expect such models tend to

provide precise information at highly resolved spatial and temporal scales, but the estimates are likely to be biased and inaccurate, particularly when the model has been configured to provide estimates at regional/national scales rather than catchment. I do understand the value and trade-offs involved in developing national-scale models as opposed to catchment-specific models, but I also think we need to be transparent about how the performance of such modelling schemes.
Apologies – this was an oversight in the original manuscript, which omitted references specifically looking at flood estimation. A paragraph has been added describing this and other more recent work (Section 3.1 para' 3). In addition, further information and references have been added regarding the operational use of G2G for flood forecasting across England, Wales and Scotland (Section 3.1 para' 2).

Of course I may not be right in this instance as my expectations are based on my own experience of using a range of conceptual and "physically based" models, and not of this G2G model. However, my difficulty here is that I could not find any evidence in the current manuscript, or in previously published papers, that the G2G model is able to provide reasonable estimates of the Q50 flood. My above recommendation for comparing the Q50 estimates derived using the SIMOBS inputs would provide clear evidence as to how well the model is performing at the national scale of interest, and these insights can be used to add defensibility, or caveats, to the conclusions drawn.
Thank you for pointing out the omission, which we hope we have now rectified.

**Reviewer 2:**
Given the previous round of feedback provided by myself and the other reviewers, I had expected a much more substantial revision in terms of clarifying data and methods, validation of G2G, and subsequent discussions. I'm afraid that this revision suggests that the authors have not grasped the magnitude of verifications needed to ground their study, how these would inform the interpretation of the results, or justify the conclusions they wish to make. I still find that the motivation of this study is well-founded but it has not been adequately executed (and the implications of their results have not been clearly communicated in the abstract or conclusions – who would benefit from this information? Does it have the potential to alter current practices?) and it may be better for the research process for me to reject the manuscript and allow the authors space to reassess their approach and perceptions of the results without being tied to the manuscript in its current form. The authors have partially addressed the need, and made a significant effort, to provide evidence that the flood characteristics modelled using baseline climate is representative of floods modelled using observed historical climate and I commend them for undertaking this effort. However, this analysis step, which I see as critical to establishing the credibility of the subsequent analyses, is somewhat concealed and is, at present, still inadequate.
We have now made a number of substantial additions, to
1. clarify data/methods/G2G evaluation (Sections 2 and 3),
2. more fully discuss the results and their implications (Sections 4 and 5), and
3. broaden the Abstract, Introduction and Conclusions (Sections 1 and  6).
We hope that these revisions are now sufficient to meet your concerns.

The data inputs used to drive the SIMOBS runs are not described under section 2 of the data, but rather briefly introduced in Section 3 of the methods. I had previously assumed that G2G was run at a daily time step given the RCM outputs were at a daily time step. (Note that details regarding the time step of the flood modelling are missing and need to be rectified. Apologies for missing this before.) However, the description of the observed data states that "Precipitation was subdivided uniformly through the day, and temperature varied sinusoidally between the extremes." suggesting that G2G was modelled at a subdaily time step for the SIMOBS runs. This is problematic because the floods modelled from SIMOBS and the baseline runs will not be comparable. Assuming a uniform

subdaily rainfall pattern (if G2G is intended for producing subdaily flood responses) introduces other errors as a uniform rainfall pattern is much more likely to result in a smaller flood response compared with a more variable rainfall temporal pattern. The G2G runs based on data that's been gridded based on observations needs to equivalent in time steps and spatial resolution to the baseline runs.

A sub-section has been added (new Section 2.1), describing the observation-based driving data. Information on the model time-step has been added (Section 3.1 para' 1). Both the SIMOBS and SIMRCM runs are done in the same way, e.g. by equally sub-dividing daily precipitation data – this has been clarified (Section 2.2). Recent work (Kay and Brown 2023) has shown that, while use of hourly, rather than equally-distributed daily, precipitation does improve the simulation of flood peaks derived from daily mean flows, it has relatively little effect on the simulated future changes in peak flows for the larger catchments (>= 50km2) included in this work. Information on this has been added (Section 3.1 para' 3). We typically do not include small catchments, because of the use of daily data (now clarified in Section 3.1 para' 1). The potential for differences in the SIMOBS and SIMRCM results to be due to differences in spatial resolution of the respective driving data has been added to the Discussion (Section 5.1 para' 4).

The authors provide references that provide evidence that the G2G hydrological model has been verified for the purpose of analysing floods across Great Britain and for flood forecasting. If these studies specifically reproduce the specific flood features of interest in this study, namely spatial extent and coherence, seasonality, frequency, and duration, then this needs to be stated and will be the validation needed for this study. If the references provided by the authors do not specifically validate the reproduction of the flood features relevant to this study, then this validation needs to be conducted by the authors here for any of the subsequent analysis to be credible. I acknowledge the challenge in doing this given the authors have pointed out the difficulty in validating G2G outputs based on observed climate data, as they have now modelled, given the lack of gridded flow data based on observed streamflow data. However, validations of gridded runoff and streamflow products are not at all without precedence, for example: Frost, A.J., Shokri, A., Keir, G., Bahramian, K., Azarnivand, A., 2020. Evaluation of the Australian Landscape Water Balance model (AWRA-L v7) (Bureau of Meteorology Technical Report) and I note that Reviewer 2 had suggested that such an evaluation could be conducted at some representative locations and this too would be sufficient.

A number of references relating to previous work evaluating G2G for flood modelling and forecasting have now been added, with further detail (Section 3.1 para's 2,3) – apologies that most were initially omitted. However, they do generally focus on evaluation against gauged flow data at a set of individual gauge locations. We do not believe that it is possible to specifically evaluate factors like spatial extent and coherence of floods in this way, given the very limited number of gauged locations (<800) relative to the number of 1km river grid cells across GB (nearly 20,000). Further barriers/complications to such an evaluation include differing periods of data availability, differing reliability of high flow gauging, and differing artificial influences on gauged flows.
Having looked at the BoM report cited above, we cannot see where this looks at evaluating spatial extent or coherence of floods (but apologies if this has been missed). It may be possible to use remotely sensed spatial datasets of, for example, soil moisture to help evaluate spatial performance of a model, but even then there are difficulties in terms of being able to compare equivalent things, as remotely sensed data typically only include moisture in the top layer of the soil, whereas G2G provides depth-integrated soil moisture for the whole soil column.
In terms of spatial extent and coherence of floods, it is the use of the underpinning spatial datasets on landscape properties that achieves this in combination with the spatial pattern of precipitation over time. The grid-based approach does not aggregate landscape properties at the scale of the gauged catchment, nor does it employ catchment-average rainfall, so the spatial extent and coherence of flooding should be better captured across the modelled domain. Figure 2 of Moore et al. (2012) provides an insightful illustration of the extent and coherence properties of G2G flood risk

assessments for a 10-year flood exceedance for an area over the English Midlands during the summer 2007 floods. Also, Moore et al. (2006) specifically considers the issue of forecasting extreme floods and the value of a distributed model such as G2G in this context, including illustrative case studies. A detailed evaluation of G2G for use in rapid response catchments (typically ungauged) is made in Cole et al. (2013). This provides evidence, across Britain over four water years and for case study storms, of good performance stratified by catchment type (area, urbanisation, headwater).

The results and discussion need to be presented with respect to the validation results. The ability of the SIMOBS to reproduce the flood features of interest will determine the caveats with which the climate change impact results are interpreted. At present, given the validation of SIMOBS in reproducing these flood features are missing, I find that the discussion and conclusions do not adequately represent the level of conjecture that should currently be ascribed.

As stated in response to similar comments from Reviewer 1, the differences in flood characteristics between the baseline climate model-driven run and observation-driven run have now been described more fully in the Results (Section 4), and a paragraph has been added to the Discussion (Section 5.1 para' 4). The need to interpret future changes "in the context of differences in event characteristics between the baseline climate projection-driven model runs and an observation-driven model run" has also now been added to the Abstract and Conclusions (Section 6 para' 2). We hope that these changes make the level of uncertainty much clearer.

I strongly encourage the authors to revisit their approach as their aim to examine the impacts of climate change on widespread flooding is a worthy endeavour. I can see that they have undertaken substantial analyses to shed light on this topic and these efforts should not be cast aside. Rather additional work is needed to properly substantiate their claims. This work contains the endings of what could be a very good paper, but the foundations need to be properly established.

Thank you for your continuing efforts to help improve our manuscript. We hope that the substantial additions now made, described above, are sufficient to justify acceptance/publication.

**Reviewer 3:**
I want to thank the authors for taking the time to carefully address the comments made during the first round of revision. It has greatly improved in my opinion.

Thank you.

Some final suggestions for technical corrections:
- Add an explanation of the percentages shown in the figures to the caption of Figure 3.

Done (moved from main text).

- Figure 4 is difficult to read. I suggest to use a histogram style similar to the one presented in Figure 5.

Done.

- x-labels are missing in Figure 9

Done.